# Mechanism of co-transcriptional cap snatching by influenza polymerase

Alexander Helmut Rotsch[1,7], Delong Li[1,5,7], Maud Dupont[2,7], Tim Krischuns[2], Ute Neef[1], Christiane Oberthür[1], Alice Stelfox[3], Maria Lukarska[3,4], Isaac Fianu[1,6], Michael Lidschreiber[1], Nadia Naffakh[2✉], Christian Dienemann[1✉], Stephen Cusack[3✉] & Patrick Cramer[1✉]

Influenza virus mRNAs are stable and competent for nuclear export and translation because they receive a 5′ cap(1) structure in a process called cap snatching[1]. During cap snatching, the viral RNA-dependent RNA polymerase (FluPol) binds to host RNA polymerase II (Pol II) and the emerging transcript[2,3]. The FluPol endonuclease then cleaves a capped RNA fragment that subsequently acts as a primer for the transcription of viral genes[4,5]. Here we present the cryogenic electron microscopy structure of FluPol bound to a transcribing Pol II in complex with the elongation factor DSIF in the pre-cleavage state. The structure shows that FluPol directly interacts with both Pol II and DSIF, positioning the FluPol endonuclease domain near the RNA exit channel of Pol II. These interactions are important for the endonuclease activity of FluPol and FluPol activity in cells. A second structure, trapped after cap snatching, shows that the cleaved capped RNA rearranges within FluPol, directing the capped RNA 3′ end toward the FluPol polymerase active site for viral transcription initiation. Together, our results provide the molecular mechanisms of co-transcriptional cap snatching by FluPol.

Influenza is an acute respiratory disease that causes 290,000 to 650,000 human deaths each year[6]. Influenza is caused by an infection with influenza A or B viruses, which circulate in temperate regions as seasonal influenza[6]. However, rare zoonotic transmissions can cause pandemic influenza outbreaks with high mortality and economic losses[7,8]. There is current concern that the unexpected susceptibility of dairy cows to avian H5N1 strains may be a path towards a new pandemic[9–11]. Influenza viruses are segmented negative-sense RNA viruses that infect the respiratory tract epithelial cells in humans[8]. After infection, the eight viral ribonucleoproteins are released into the cytoplasm and imported into the nucleus, where transcription of viral genes into mRNA and replication of the viral genome occur[12,13]. Each viral ribonucleoprotein contains a genome segment that is encapsidated by multiple copies of the viral nucleoprotein and one copy of the viral RNA-dependent RNA polymerase (FluPol). FluPol consists of subunits PA, PB1 and PB2 and has been structurally characterized[2,14,15].

Viral transcripts must contain a 5′ cap structure and a 3′ poly(A) tail to ensure stability, nuclear export and efficient translation[16]. However, unlike non-segmented negative-sense RNA viruses, the influenza virus genome does not encode enzymes that synthesize a 5′ cap[17]. Instead, FluPol utilizes capped RNA primers that are cleaved from nascent host transcripts in a process called cap snatching[1,5]. The FluPol PB2 cap-binding domain binds a nascent 5′ capped host RNA, and the PA endonuclease domain cleaves off 10–15 nucleotides (nt) from the 5′ end. The 3′-terminal nucleotides of this RNA primer then anneal to the 3′ end of the viral genome segment and prime transcription of the viral mRNA[14,18,19].

Capped host transcripts are synthesized by cellular Pol II. Pol II transcription starts with assembling a pre-initiation complex consisting of Pol II and the general transcription factors at gene promoters[20]. To escape from the gene promoter, the largest Pol II subunit RPB1 C-terminal domain (CTD) heptad repeats are phosphorylated at serine 5 and serine 7 by the TFIIH CDK-activating kinase (CAK)[21,22]. CTD phosphorylation and the growing nascent RNA transcript cause the initiation factors to dissociate from Pol II[22,23]. Recruitment of the elongation factor DSIF after synthesis of around 20 nt of RNA establishes the early Pol II elongation complex (Pol II–DSIF). This complex is then converted to a paused elongation complex (PEC) containing the negative elongation factor NELF at a transcript length of 25–50 nt (refs. 23–25). Synthesis of the 5′ cap occurs co-transcriptionally by the capping enzymes RNGTT, RNMT and CMTR1 (ref. 25) in the context of the Pol II–DSIF elongation complex or the PEC. RNGTT is a bifunctional enzyme that acts as a triphosphatase and guanylyltransferase, creating a GpppN structure at the 5′ end of the Pol II transcript. RNMT and CMTR1 are methyltransferases that add a methyl group to N7 of the cap guanosine and the 2′-OH of the first regular nucleotide, respectively, producing the m7GpppmN cap(1) structure[25], which the cap-binding domain of FluPol subunit PB2 tightly binds during cap snatching[26,27].

Cap snatching depends on host transcription, as it has been shown that inhibition of Pol II using α-amanitin impairs viral replication[3].

[1]Department of Molecular Biology, Max Planck Institute for Multidisciplinary Sciences, Goettingen, Germany. [2]RNA Biology and Influenza Viruses, Institut Pasteur, Université Paris Cité, CNRS UMR3569, Paris, France. [3]European Molecular Biology Laboratory, Grenoble, France. [4]Department of Molecular and Cell Biology, University of California, Berkeley, CA, USA. [5]Present address: Mechanisms of Cellular Quality Control, Max-Planck-Institute of Biophysics, Frankfurt, Germany. [6]Present address: Division of Chemistry and Chemical Engineering, California Institute of Technology, Pasadena, CA, USA. [7]These authors contributed equally: Alexander Helmut Rotsch, Delong Li, Maud Dupont. ✉e-mail: nadia.naffakh@pasteur.fr; christian.dienemann@mpinat.mpg.de; cusack@embl.fr; patrick.cramer@mpinat.mpg.de

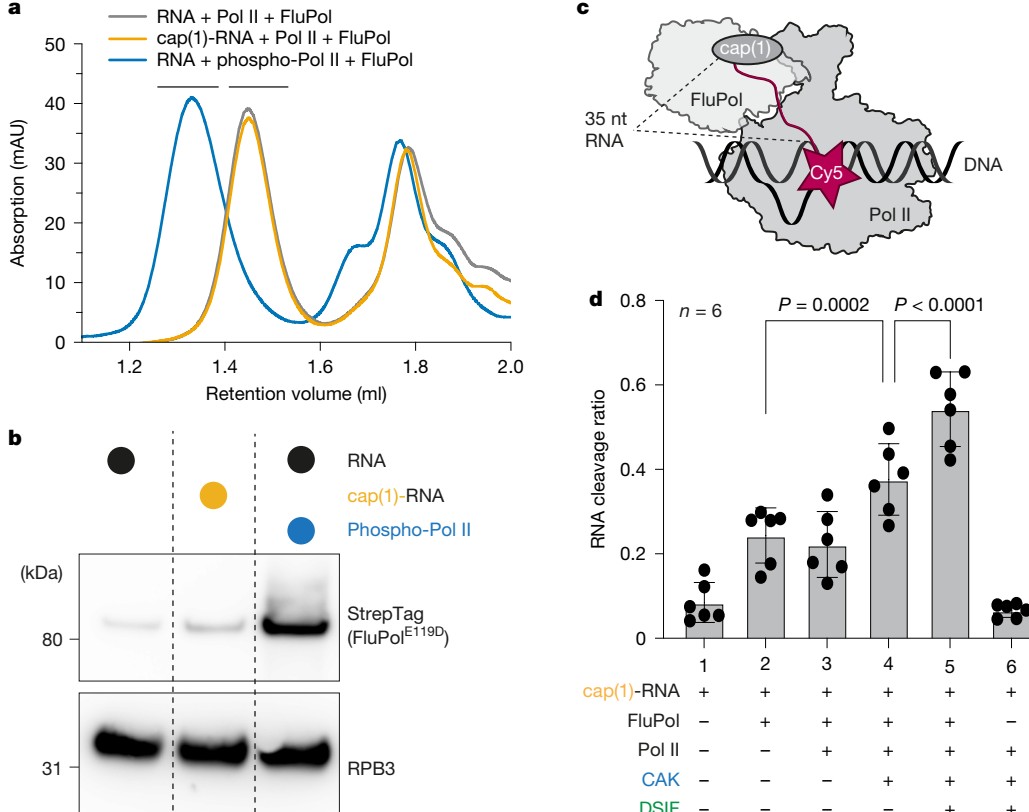

**Fig. 1 | FluPol recognizes the Pol II elongation complex. a**, Absorbance at 280 nm of analytical SEC runs of Pol II elongation complex containing a 35-nt RNA with or without cap(1) and with or without CAK phosphorylation, and with FluPol[E119D]. Different colours represent different chromatography runs. Black bars above the chromatogram depict Pol II complex fractions that were analysed by Western blot in **b**. **b**, Western blot of Pol II containing peak fractions stained against RPB3 (Pol II) and Twin-StrepTag (FluPol subunit PB2). Individual lanes represent different SEC runs. **c**, Schematic of the endonuclease cleavage assay. The cap(1)-RNA is Cy5-labelled on the 3′ end. **d**, The fraction of cleaved RNA (intensity of cleaved product divided by intensity of all bands, see Extended Data Fig. 1c) depends on the factors added. Each point reflects one experimental replicate (*n* = 6); mean ± s.d. *P* values were calculated using a linear mixed-effects model (substrate as a fixed effect, experimental replicate as a random effect, two-sided, no multiple testing correction).

FluPol localizes primarily at the 5′ end of host genes and associates with the Pol II CTD that is phosphorylated at serine 5 residues, indicating that cap snatching occurs during early phases of Pol II transcription[2,28–30]. Cell-based protein–protein interaction assays indicate that FluPol binds not only to the CTD but also to the Pol II body[31]. Co-immunoprecipitation–mass spectrometry experiments have shown that the elongation factor DSIF co-purifies with FluPol[5,32], and other studies suggest that FluPol depends on the cap(1) structure for cap snatching[26]. However, how FluPol interacts with the host transcription machinery for cap snatching at the molecular level is unknown.

Here we show that FluPol binds to the transcribing Pol II–DSIF complex for efficient cap snatching. Furthermore, we report two cryogenic electron microscopy (cryo-EM) structures of FluPol bound to a Pol II–DSIF elongation complex before and after endonucleolytic RNA cleavage by FluPol. The structures show that during cap snatching, the PA endonuclease domain of FluPol binds near the RNA exit channel of Pol II and that this interaction is stabilized by DSIF. Furthermore, using cell-based minigenome assays, we confirm that mutation of residues forming the interface between FluPol and the Pol II–DSIF elongation complex reduces FluPol activity in cells. In summary, we present the molecular mechanism of cap snatching by FluPol.

## FluPol snatches cap from Pol II elongation complex

To study the molecular basis of cap snatching, we first investigated how the formation of a complex between FluPol and transcribing Pol II (Pol II elongation complex) depends on the cap(1) structure and CTD

phosphorylation. We purified *Sus scrofa* Pol II (99.9% sequence identity to human Pol II, with 4 amino acid differences) from the endogenous source[33]. Whereas in preliminary studies reconstituting the cap-snatching complex[30], we used bat FluPol(H17N10), here we used recombinant, viral promoter-bound FluPol from the influenza strain A/Zhejiang/DTID-ZJU01/2013(H7N9)[34,35] (Extended Data Fig. 1a). To reduce RNA cleavage and enhance complex stability, we used the PA(E119D) mutant of FluPol (FluPol[E119D]; Extended Data Fig. 3j), which has impaired endonuclease activity[36,37]. A Pol II elongation complex containing a 35-nt cap(1)-RNA, 45-nt template, and non-template DNA was assembled as established previously[38]. The 35-nt RNA length was chosen considering a 12-nt RNA primer produced by cap snatching[19,39], an additional 3 nt bound by the PA endonuclease[37], and 20 nt RNA bound within the Pol II elongation complex[33].

We next monitored binding of FluPol[E119D] to the Pol II elongation complex by size-exclusion chromatography (SEC) using unmodified RNA and Pol II, cap(1)-RNA, or Pol II that was phosphorylated with CAK. Without CTD phosphorylation and a cap(1) structure, co-elution of FluPol with Pol II could barely be detected (Fig. 1a,b). When a cap(1)-modified RNA was used, the signal for FluPol in the Pol II containing peak slightly increased (Fig. 1b). However, when the Pol II CTD was phosphorylated by CAK, the amount of FluPol associated with Pol II in the peak fractions strongly increased (Fig. 1b). Additionally, the elution volume of the complex peak shifted towards higher molecular weight, indicating the formation of a stable complex (Fig. 1a). Thus, the addition of a cap(1) structure to the RNA has a negligible effect on the interaction between FluPol and the Pol II elongation complex. By contrast, phosphorylation

of the Pol II CTD is the main determinant for the recruitment of FluPol to a Pol II elongation complex, consistent with in vivo data demonstrating the importance of the Pol II CTD for viral transcription[2,29].

We next tested whether the increased affinity of FluPol[E119D] to Pol II by CTD phosphorylation also results in enhanced endonuclease activity by wild-type FluPol. To monitor RNA cleavage, we developed a fluorescence-based assay using in vitro-capped RNA labelled with Cy5 at the 3′ end (Fig. 1c). We assembled Pol II elongation complexes in vitro essentially as described above, added wild-type FluPol and then visualized the cleaved 3′ end of the RNA that remains attached to Pol II by denaturing PAGE. The primary cleavage product detected was 20–25 nt long, corresponding to the expected 10- to 15-nt primer generated by FluPol (Extended Data Fig. 1b). Additionally, small amounts of an additional cleavage product of around 30 nt were produced. Comparing the different RNA substrates, we did not observe an increase in RNA cleavage by FluPol in the context of a Pol II elongation complex compared to free RNA (Fig. 1d, Extended Data Fig. 1b,c and Supplementary Table 1). However, RNA cleavage increased when we phosphorylated the CTD of Pol II by adding CAK (Fig. 1d and Extended Data Fig. 1b), in line with previous reports[40]. This suggests that CTD phosphorylation enhances recruitment of FluPol to Pol II and stimulates cleavage of RNA that is bound to Pol II.

Next, we tested whether the presence of the elongation factor DSIF, which binds Pol II during early elongation, stimulates the cleavage of Pol II-bound RNA. Indeed, cleavage of Pol II-bound RNA was stimulated ~1.5-fold when DSIF was added in excess to the Pol II (Fig. 1d and Extended Data Fig. 1b, c). Finally, we tested whether wild-type FluPol can extend the snatched RNA primer using a radioactive FluPol RNA extension assay. We found that FluPol alone can extend the cleaved RNA fragments to some degree. Furthermore, FluPol-dependent RNA extension increases in the presence of a Pol II–DSIF elongation complex, which indicates that the more efficient endonuclease reaction provides more usable RNA primers for FluPol transcription (Extended Data Fig. 1d).

In summary, the cap(1) structure only has a minor effect on FluPol binding to Pol II, whereas CTD phosphorylation by CAK strongly enhances FluPol recruitment and stimulates cleavage of Pol II-bound RNA by FluPol to some extent. Additionally, DSIF, when added to the Pol II elongation complex, significantly enhances RNA cleavage further, suggesting that DSIF is part of the Pol II complex that is recognized by FluPol. Moreover, we have demonstrated that the RNA emerging from the Pol II–DSIF elongation complex can be used to prime RNA synthesis by FluPol. Thus, we conclude that FluPol recognizes the phosphorylated Pol II–DSIF elongation complex as a minimal substrate for efficient cap snatching.

## FluPol–Pol II–DSIF complex structure

After determining the components required for efficient cap snatching by FluPol in vitro, we next sought to structurally characterize a cap-snatching complex comprising FluPol, Pol II, DSIF and capped RNA by cryo-EM. To that end, we first assembled a Pol II–DSIF elongation complex containing a 35-nt cap(1)-RNA in the presence of the CAK and ATP to phosphorylate the Pol II CTD. To capture the normally transient cap-snatching complex prior to RNA cleavage, we added FluPol[E119D] at low Mg[2+] concentration, a condition in which cleavage is minimal (Extended Data Fig. 1e). The complex was purified and stabilized using GraFix[41] prior to cryo-EM sample preparation (Extended Data Fig. 2a). Cryo-EM data acquisition yielded 6,423,874 particles that were further sorted by 3D classification, which yielded a subset of 369,858 particles that show good density for the Pol II–DSIF elongation complex, as well as FluPol resolved at 3.3 Å overall resolution (Extended Data Fig. 2b–h and Extended Data Table 1). From this consensus refinement, we performed focused refinements of FluPol and the Pol II–DSIF elongation complex (with resolutions of 2.90 Å and 2.94 Å, respectively), which

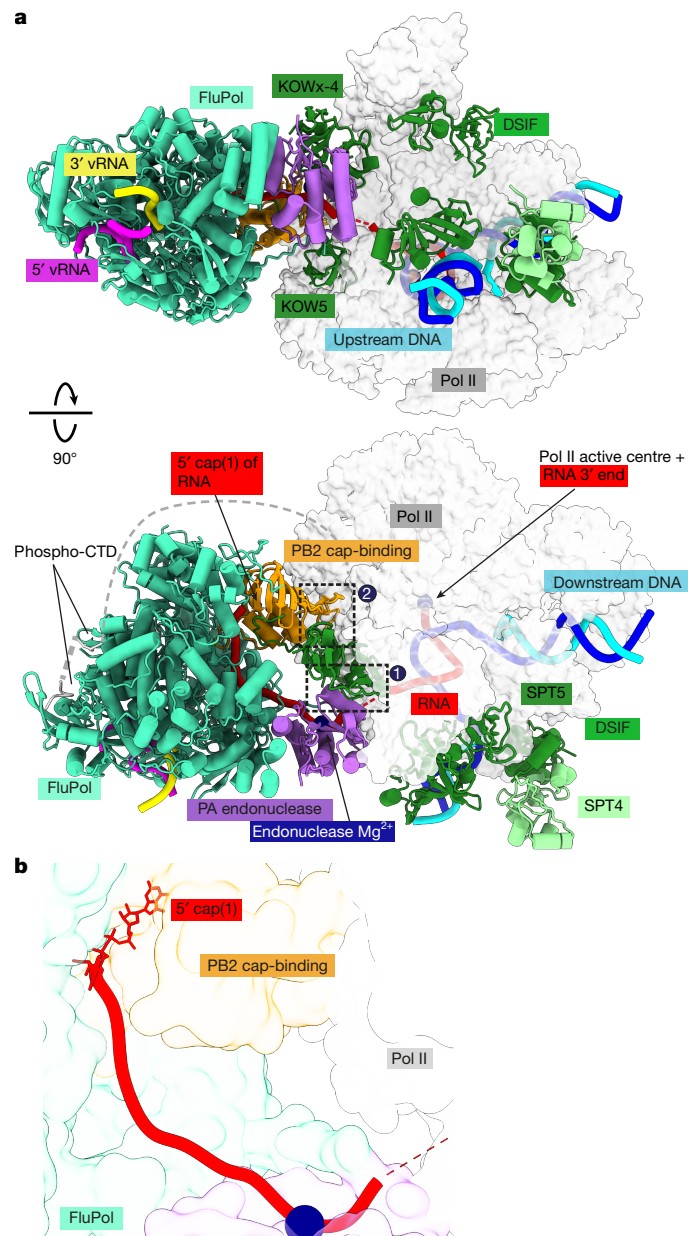

**Fig. 2 | Structure of the pre-cleavage cap-snatching complex. a**, Two views of the overall structure of the pre-cleavage FluPol–Pol II–DSIF elongation complex complex in cartoon representation except Pol II, which is shown as surface. Dashed black boxes represent the locations of the two interfaces shown in Fig. 3a–c. The structure is shown in a FluPol side view and Pol II top view (top) as well as front view of FluPol and side view of Pol II (bottom). **b**, The RNA path within FluPol. Proteins are shown as transparent surfaces. The RNA is shown as ribbon tracing of the backbone. Parts of the FluPol model were removed for clarity.

enabled us to build and refine an atomic model for the complete cap-snatching complex (Fig. 2a).

The structure shows that FluPol binds to the Pol II–DSIF elongation complex near the RNA exit channel of Pol II (Fig. 2a). The PA endonuclease of FluPol interacts with the KOWx-4 domain of DSIF that forms a clamp around the exiting RNA in the absence of FluPol[33] (Fig. 2a, interface 1). In the complex, KOWx-4 is rotated approximately 180° around its longitudinal axis and shifted by around 22 Å compared with the Pol II–DSIF elongation complex structure[33], and the Pol II stalk containing subunits RPB4 and RPB7 is also repositioned (Extended Data

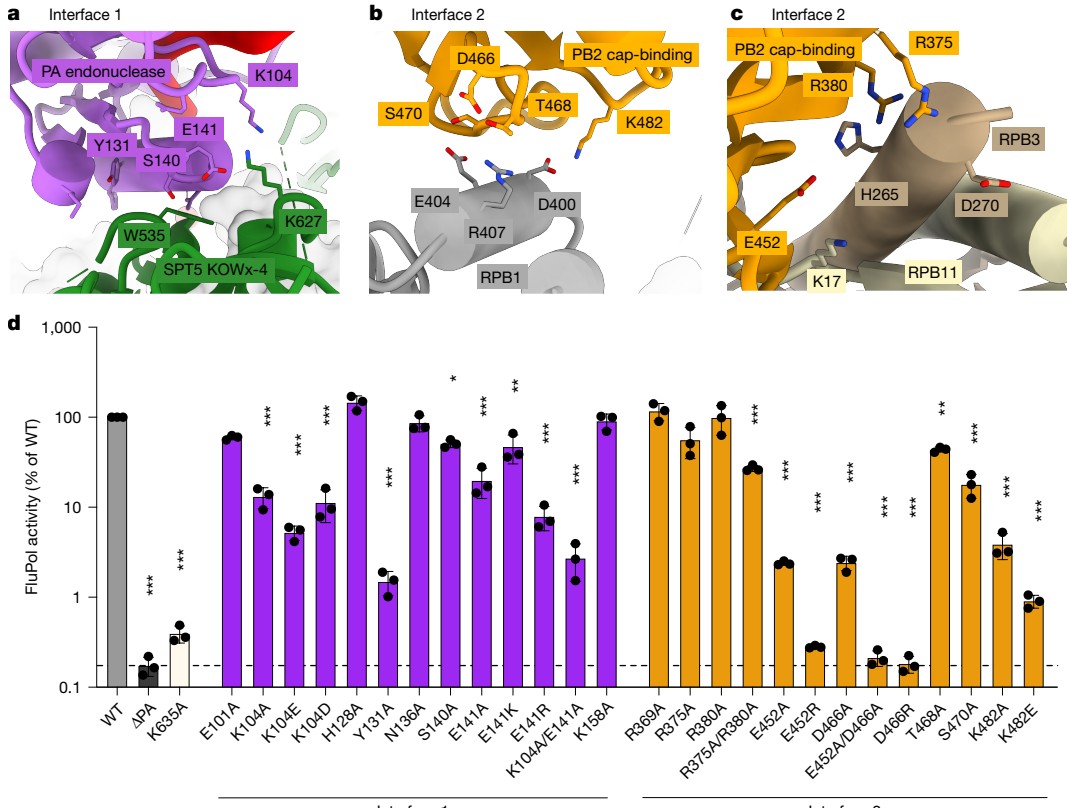

**Fig. 3 | New FluPol–Pol II–DSIF elongation complex interfaces. a–c**, Zoom-ins on the interfaces between the FluPol PA endonuclease domain and the DSIF KOWx-4 domain (**a**), the FluPol PB2 cap-binding domain and RPB1 (**b**) or RPB3 and RPB11 (**c**). Mutated amino acids mutated are shown in stick representation and coloured by heteroatoms if the mutation reduced FluPol activity significantly in a cell-based minigenome assay. **d**, Cell-based minigenome assay of A/WSN/33 FluPol activity for the indicated PA and PB2 mutants. HEK-293T cells were co-transfected with plasmids encoding PB2, PB1, PA and NP with a model vRNA encoding firefly luciferase. Luminescence was normalized to a transfection control and is represented as percentage of wild-type (WT) FluPol. The K635A mutant was used as a transcription-defective control. The dotted line represents background signal as measured in the absence of the PA subunit (ΔPA). Each point reflects one biological replicate ($n = 3$), depicted as mean ± s.d., ***$P < 0.001$, one-way ANOVA on log-transformed data with Dunnett's multiple comparisons test referenced to wild-type FluPol.

Fig. 3a,b). The PB2 cap-binding domain of FluPol inserts between the Pol II subunits RPB1, RPB3 and RPB11 to bind the Pol II dock domain, which is located below the RNA exit channel of Pol II (Fig. 2a, interface 2). In line with our observation that FluPol recruitment to Pol II strongly depends on CTD phosphorylation, we observe density for serine 5 phosphorylated CTD residues in the two previously reported CTD binding sites of FluPol[2,42,43] (Extended Data Fig. 3c–f).

We could trace continuous density for most of the RNA from the capped 5′ end in the PB2 cap-binding domain of FluPol all the way to the 3′ end located in the Pol II active site (Fig. 2a,b and Extended Data Fig. 3g,i). This confirms that we successfully resolved the cap-snatching complex prior to endonuclease cleavage. Therefore, we called this structure the pre-cleavage complex. The cap(1) and the first four nucleotides of the RNA are well ordered and tightly bound to the PB2 cap-binding and midlink domains, as observed before[18,44]. The methylated 2′ OH of the first transcribed base packs against I260 from the PB2 midlink domain (Extended Data Fig. 3h,i), as proposed previously[26]. The interaction of FluPol with the cap(1) structure is supported by parts of a previously unresolved linker between the KOWx-4 and KOW5 domains of DSIF (SPT5 residues 647–703), which interacts directly with the RNA 5′ end and the cap-binding domain of PB2 (Extended Data Fig. 3h,i). Phosphorylation of serine residues in this linker has been reported to be involved in pause release[45]. Deleting this linker or alanine mutants of two of the potential phenylalanines that might contact the cap structure only slightly decreases the endonuclease activity of FluPol in vitro (Extended Data Fig. 3k,l).

The nucleotides between the cap-binding domain and the FluPol endonuclease could only be resolved at low resolution (Extended Data Fig. 3g,i), probably owing to the flexibility of this RNA region. This precluded identification of the exact sequence register, although structural modelling (Methods) allows for 9–15 nt of RNA to be placed between the endonuclease and the cap-binding domains (Fig. 2b), in agreement with the primer lengths of 10–15 nt that are produced by co-transcriptional cap snatching in vivo[19,39]. In summary, we visualized the structure of a pre-cleavage state of FluPol bound to transcribing Pol II during cap snatching, explaining how DSIF stimulates cleavage of Pol II-bound RNA.

## Effect of Pol II binding on FluPol in vivo

The biochemical analysis of FluPol endonuclease activity and the structure of the pre-cleavage complex show that FluPol binds the Pol II–DSIF elongation complex, and that the interaction between FluPol and DSIF is important for cap snatching in vitro. Next, we investigated whether the observed interactions between FluPol and the Pol II–DSIF elongation complex are also required for FluPol activity in vivo—that is, in a cellular context. For this purpose, we utilized the pre-cleavage complex structure to identify 16 FluPol residues at the interface with the Pol II–DSIF elongation complex that show high conservation across various influenza strains (Extended Data Fig. 4a,b). We then used a luciferase-based minigenome assay to test FluPol activity in cells after mutating interface residues between PA and DSIF (Fig. 3a, Interface 1), as well

as PB2 and Pol II (Fig. 3b,c, Interface 2), to alanines or to their reverse charge counterpart (Supplementary Table 2). As a positive control for defective FluPol transcription, we used the CTD binding mutant PA(K635A)[42]. Before investigating FluPol activity of these mutants in the minigenome assay, we ensured that all variants are expressed at a similar level as in wild-type FluPol (Extended Data Fig. 5a,b). Of the 26 mutants tested, 19 show a significant reduction in FluPol activity (Fig. 3d and Supplementary Table 3).

Mutation of PA residues K104 or E141 to alanine reduced FluPol activity tenfold in the minigenome assay (Fig. 3d). Both residues are close to the conserved SPT5 residue K627 (Extended Data Fig. 4c), with which PA E141 and K104 might form a salt bridge network (Fig. 3a). Individual charge-reversal mutants of K104 and E141 did not further decrease FluPol activity, but combining both alanine mutations did (Fig. 3d). The side chain geometry in this interface is likely to be flexible enough to rearrange in order to compensate for single mutations of the surface charge. Additionally, mutating PA residue Y131, which might be involved in a hydrophobic interaction with SPT5 residue W535, shows the most substantial reduction of luciferase activity of all mutations tested on this surface. PB2 residues D466, T468, S470 and K482 are located at interface 2 between the PB2 cap-binding domain and the RPB1 dock domain (Fig. 3b) and can form hydrogen bonds, as well as salt bridges, with RPB1 residues D400, E404 and R407. From this interface, the T468A and S470A mutants retained around 40% and 20% of wild-type activity, respectively, whereas the other alanine mutations reduced FluPol activity to less than 10% of wild-type activity (Fig. 3d). The charge-reversal mutants D466R and K482E reduce FluPol activity more than the alanine variants. Furthermore, mutation of PB2 residue E452, which is involved in a salt bridge with K17 of RPB11 (Fig. 3c), also reduced FluPol activity in vivo (Fig. 3d). When combined with PB2 D466A, FluPol activity is reduced to a background level. Whereas individual mutations of PB2 R375 or R380—residues involved in the interaction with the RPB3 C-terminal residues H265 and D270—to alanine do not alter FluPol activity, mutating both residues significantly decreases FluPol activity (Fig. 3d). Additionally, the interface residues in RPB1, RPB3 and RPB11 are highly conserved between mammals and birds (Extended Data Fig. 4d–f). These results show that the interface between the PB2 cap-binding domain and the Pol II surface is important for FluPol activity in vivo.

Together, the results show that the integrity of the PA endonuclease interface with DSIF, as well as the interface between PB2 and Pol II, are vital for efficient FluPol activity in vivo. This agrees with our biochemical and structural data showing that the Pol II–DSIF elongation complex is the substrate for cap snatching by FluPol (Fig. 1d).

## Pol II interface affects transcription

Mutations at interfaces 1 and 2 reduce FluPol activity in cells; however, this effect could be caused by defects in viral transcription and replication. To determine whether the effects of the mutations are transcription-specific, we performed strand-specific quantitative PCR with reverse transcription (RT–qPCR) in the context of the minigenome assay (Extended Data Fig. 5c and Supplementary Table 4). We then calculated the ratio of influenza mRNA over viral RNA (vRNA) to determine which FluPol mutations specifically affect viral transcription[2,46] (Extended Data Fig. 5d). Mutations at the PA–DSIF interface (PA(Y131A) and PA(K104A/E141A)) did not specifically reduce the mRNA/vRNA ratio, suggesting that these residues might be primarily important for replication in vivo. At interface 2, individual alanine mutations of PB2 E452, D466 and K482 did not affect mRNA/vRNA ratios. However, the E452R and D466R charge-reversal mutations, as well as the E452A/D466A double mutation in PB2 led to a reduced mRNA/vRNA ratio (Extended Data Fig. 5d). This shows that PB2 residues E452 and D466 are specifically required for FluPol transcription in vivo. The importance of interface 2 for viral viability was further confirmed

## Table 1 | Phenotypic and genotypic characterization of recombinant viruses

| Virus | Titre[a] (PFU ml[−1]) | Plaque diameter[b] (mm) | Mutation[c] |
|---|---|---|---|
| Wild type | $(5 \pm 1.8) \times 10^7$ | $5.82 \pm 0.68$ ($n = 51$) | |
| PA(Y131A) | $(3 \pm 0.33) \times 10^3$ | $2.92 \pm 0.77$ ($n = 84$) | PA(Y131A) (>99%) |
| PA(K104A/E141A) | $(9 \pm 0.62) \times 10^7$ | $4.37 \pm 0.72$ ($n = 27$) | PA(K104A) (>99%) PA(E141A) (99%) |
| PB2(E452R) | $(2 \pm 0.37) \times 10^7$ | $4.55 \pm 0.75$ ($n = 21$) | PB2(E452R) (>99%) PB2(S453P) (93%) |
| PB2(D466R) | $(4 \pm 0.28) \times 10^7$ | $5.72 \pm 0.86$ ($n = 40$) | PB2(D466R) (11%) PB2(D466C) (79%) PA(M211V) (27%) |
| PB2(K482E) | $(1 \pm 0.4) \times 10^6$ | $4.48 \pm 0.66$ ($n = 30$) | PB2(K482E) (99%) M2(E75K) (12%) |

[a]Determined using a plaque assay on MDCK cells (mean ± s.d. of technical triplicates). [b]Determined using a plaque assay on MDCK cells (mean ± s.d. from 21 to 84 plaques, as indicated). [c]Determined by Illumina sequencing after RT–PCR amplification of the whole viral genome. Only mutations that differ from the wild-type sequence and were found in more than 10% of the reads are indicated.

by plaque formation assay showing reduced viral titres and plaque diameter for mutants E452R and K482E (Extended Data Fig. 5e, Table 1 and Supplementary Table 5). Viruses expressing PB2(E452A/D466A) could not be rescued by reverse genetics, and viruses with the PB2 D466R mutation acquired a second site mutation R>C (Table 1 and Supplementary Table 6), further demonstrating the importance of this interface for virus viability.

In summary, the PA–DSIF interface mutations do not lead to a transcription-specific FluPol defect in vivo, whereas the PB2–Pol II interface specifically affects FluPol transcription in vivo, highlighting its importance for viral transcription.

## DSIF interface is important for endonuclease activity

Both interfaces between FluPol and the Pol II–DSIF elongation complex are required for FluPol activity in vivo, however, we could only find mutations leading to FluPol transcription-specific defects in interface 2. Since a primary replication defect may mask deficiencies in cap snatching and the endonuclease activity of FluPol, we sought to investigate the effect of these interface mutations in vitro. We purified FluPol and DSIF variants with mutations that lead to FluPol activity defects in vivo, and tested them in an endonuclease activity assay using RNA bound to a Pol II–DSIF elongation complex. All FluPol mutants tested here were validated to cleave free RNA with similar efficiency, confirming that the differences observed originate from alterations in interface 1 and 2 (Extended Data Fig. 6c,f and Supplementary Table 1).

Mutations of PA residues Y131, K104 and E141 to alanines lead to a reduction of endonuclease cleavage in the context of a Pol II–DSIF elongation complex (Extended Data Fig. 6a,b). Additionally, when the KOWx-4 domain of DSIF is deleted, endonuclease activity by FluPol is reduced to levels similar to those in a reaction without DSIF (Extended Data Fig. 3k,l). Mutating the potential salt bridge partner in SPT5, K627 (Fig. 3a), to alanine does not significantly reduce FluPol endonuclease activity, whereas an alanine mutation of the SPT5 hydrophobic surface residue W535 leads to reduced RNA cleavage by FluPol. Other interface 1 mutations tested did not alter endonuclease activity (Extended Data Fig. 6b), which is in line with their smaller effect on FluPol activity in vivo (Fig. 3d). This shows that interface 1 is important for efficient endonuclease cleavage. This is additionally confirmed by the reduced plaque diameter of viruses carrying the Y131A mutation or K104/E141A double mutation in PA (Extended Data Fig. 5e and Table 1). Although mutations in PB2 specifically affected FluPol transcription in vivo, these mutations did not alter the endonuclease activity in vitro (Extended

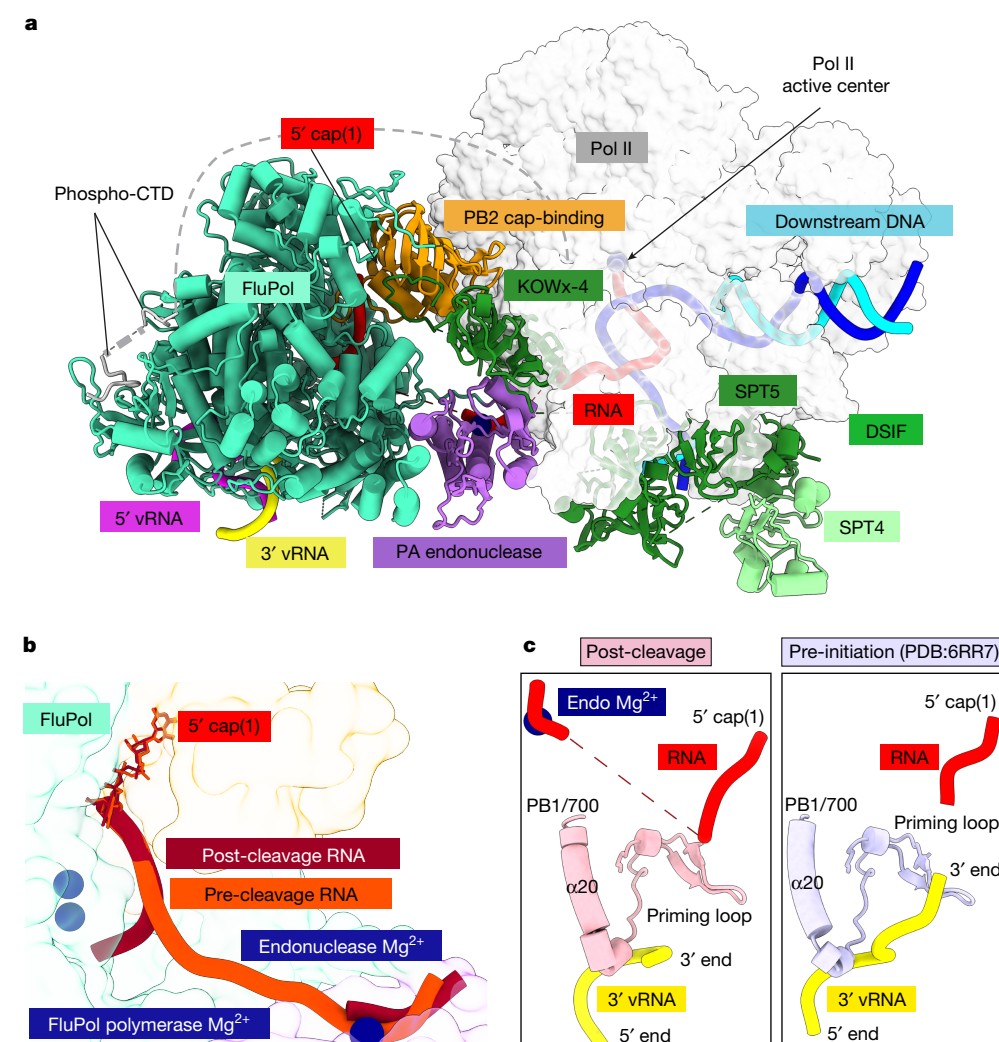

**Fig. 4 | Structure of the post-cleavage cap-snatching complex. a**, Overall structure of the post-cleavage FluPol–Pol II–DISF elongation complex in cartoon representation except for Pol II, which is shown as surface. **b**, Comparison of the RNA path in FluPol between pre and post-cleavage complex. Proteins are shown as transparent surfaces and the RNA is shown as ribbon tracing of the backbone. FluPol polymerase active site $Mg^{2+}$ atoms are modelled on the basis of the FluPol elongation complex[14]. Parts of FluPol were removed for clarity. **c**, Comparison of the FluPol polymerase active site conformations in the post-cleavage (pink) and pre-initiation (Protein Data Bank (PDB): 6RR7, light purple[44]) states. Only the priming loop, the viral mRNA and the 3′ vRNA are shown.

Data Fig. 6d,e). This is consistent with the observation that unphosphorylated Pol II alone does not directly stimulate FluPol endonuclease activity (Fig. 1d).

In summary, although PA–DSIF interface mutants do not specifically affect transcription of viral mRNA in vivo, the endonuclease activity of FluPol is reduced for these mutants. This suggests that the stability of the PA–DSIF interface is important for efficient endonuclease cleavage.

## Transition to FluPol pre-initiation

The pre-cleavage complex structure reveals the RNA trajectory directly from the cap-binding domain to the endonuclease domain of FluPol. After endonuclease cleavage of the RNA, the newly generated RNA 3′ end must be directed into the FluPol PB1 polymerase active site for RNA extension. It is also unclear whether FluPol stays attached to Pol II after RNA cleavage. To investigate these, we sought to resolve a cap-snatching complex of FluPol bound to the Pol II–DSIF elongation complex under conditions in which the PA endonuclease can cleave the RNA.

To achieve this, we assembled the Pol II–DSIF elongation complex with cap(1)-RNA and FluPol[E119D] as before, but in the presence of 3 mM

$Mg^{2+}$ (Extended Data Fig. 1e), which led to RNA cleavage during cryo-EM sample preparation (Extended Data Fig. 7a). We then performed cryo-EM as described for the pre-cleavage complex and identified a subset of particles containing cryo-EM density for FluPol and the Pol II–DSIF elongation complex (Fig. 4a and Extended Data Fig. 7b–h). The resulting post-cleavage structure is very similar to the pre-cleavage complex (Extended Data Fig. 8a), except for the path taken by the primer RNA. In particular, we could only trace the RNA from the Pol II active site until the PA endonuclease active site, after which the density discontinues abruptly (Extended Data Fig. 8b–d). This observation suggests that FluPol can stay attached to the Pol II–DSIF elongation complex despite the break in the RNA. The 5′ cap(1) structure remains bound as before in the FluPol cap-binding site. However, the cleaved RNA 3′ end points towards the FluPol polymerase active site, guided by the positive surface charge at the RNA exit channel (PB1 R260, PB2 K214 and R216) (Fig. 4b and Extended Data Fig. 8d,e). The endonuclease cleaves a fragment of 10–15 nt from the Pol II transcript[19,39], of which we can observe cryo-EM density for the first 7 nt of the primer in the post-cleavage complex, indicating that the missing nucleotides are disordered. Furthermore, in the post-cleavage complex, the priming loop near the FluPol polymerase active site is still extended and ordered

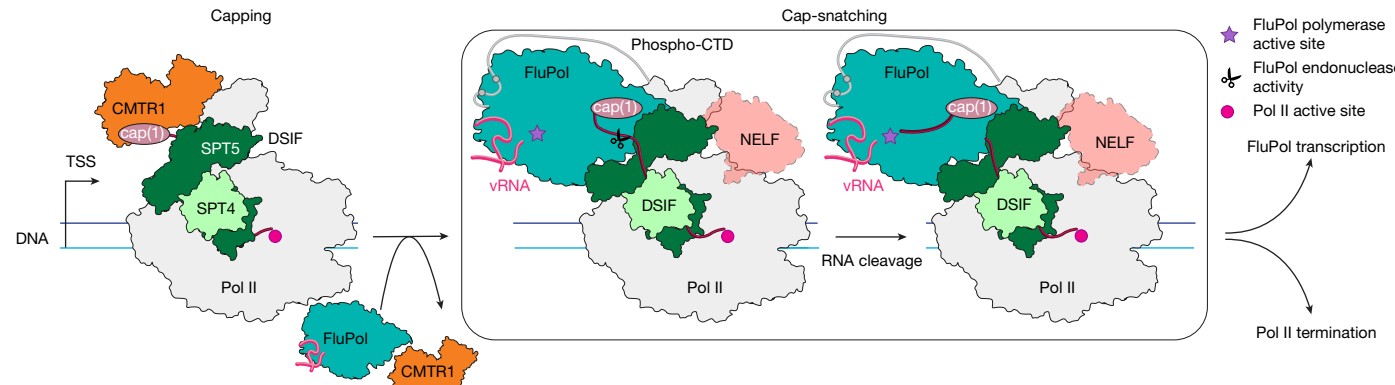

**Fig. 5 | Model of co-transcriptional cap snatching.** Capping enzymes, including CMTR1, synthesize the cap(1) structure on the RNA co-transcriptionally. After capping is finished, CMTR1 dissociates from Pol II. The resulting Pol II–DSIF elongation complex with a capped RNA is a substrate for cap snatching and is bound by FluPol. Then, the FluPol endonuclease cleaves the RNA, and FluPol may dissociate from the Pol II elongation complex surface and initiates viral transcription, whereas Pol II gets terminated.

(Fig. 4c and Extended Data Fig. 9a), showing that the snatched RNA primer has not yet base paired with the viral RNA template 3′ end. Thus, FluPol in the post-cleavage cap-snatching state resembles very closely the FluPol pre-initiation complex previously reported[18,47] (Extended Data Fig. 9b). Rotation of the cap-binding domain after primer cleavage, therefore, does not seem to be required to direct the primer into the polymerase active site as previously proposed[48]. Instead, the flexible cap-binding domain is probably fixed in the pre-initiation orientation after binding the Pol II–DSIF elongation complex in the pre-cleavage state (Extended Data Fig. 9c).

In summary, RNA cleavage by the FluPol endonuclease results in a new trajectory of the capped primer that is indicative of a FluPol pre-initiation complex. Thus, the PA endonuclease activity on Pol II-bound capped RNA leads to a state of FluPol that is ready to initiate viral transcription with minimal conformational changes.

## Discussion

The results presented here close a major gap in our understanding of the life cycle of one of the most common human viral pathogens. By combining structural, biochemical and cellular approaches, we propose a molecular mechanism of cap snatching by FluPol that involves three major steps (Fig. 5 and Supplementary Video 1). First, FluPol directly binds to the host transcription machinery. The minimal substrate for efficient cap snatching is a Pol II–DSIF elongation complex with a cap(1)-RNA and a phosphorylated Pol II CTD, which is found during early host transcription[23,24,33]. Second, FluPol endonuclease cleaves the RNA, generating a 10- to 15-nt primer. After cleavage, the new 3′ end of the capped RNA primer is directed towards the FluPol polymerase active site, resulting in a conformation that closely resembles a FluPol pre-initiation complex. Third, the 3′ end of the capped RNA primer can anneal to the vRNA template for viral mRNA synthesis.

Building on our mechanistic understanding of co-transcriptional cap snatching, our model also provides insights into how cap snatching is coordinated with host transcription by Pol II (Fig. 5). Several early Pol II transcription states have been structurally characterized, including promoter-proximal pausing[49], RNA capping[25], pause release into processive elongation (EC*)[50] and premature termination[51]. Comparison of the cap-snatching complex structure with that of the 2′-OH methyltransferase CMTR1 bound to the Pol II–DSIF elongation complex[25] shows that binding of FluPol and CMTR1 to Pol II is mutually exclusive (Extended Data Fig. 9e). Since CMTR1 is essential for cap snatching to occur[26], it is likely that CMTR1 has to dissociate after the 2′-OH methylation that completes cap(1) synthesis. CMTR1 dissociation from the Pol II–DSIF elongation complex then allows for FluPol binding to the completed cap and the Pol II–DSIF surface, as observed in the pre-cleavage structure (Fig. 2a). NELF binding to the Pol II–DSIF elongation complex establishes the PEC[49], which can accommodate FluPol binding without clashes (Extended Data Fig. 9f). However, NELF-E is thought to assist recruitment of the nuclear cap-binding complex to the completed cap[52,53]. The exact role of the cap-binding complex in cap snatching is not yet understood.

Active elongation in the EC*, in which PAF1c and SPT6 are bound to Pol II, and termination factors such as Integrator or XRN2, however, are sterically incompatible with simultaneous FluPol binding (Extended Data Fig. 9g–i). Thus, the window of opportunity for cap snatching probably opens during Pol II early elongation (Pol II–DSIF elongation complex) and pausing (PEC), and closes after pause release and formation of the EC* or premature termination. Within that window of opportunity, early elongating and paused Pol II represent relatively long-lived substrates for cap snatching as Pol II resides in this phase for several minutes[54,55]. FluPol binding to the KOWx-4-KOW5 linker of SPT5 might further extend the residence time of the paused Pol II by preventing phosphorylation of the linker, which was shown to be important for pause release[45].

The exact fate of FluPol after cap snatching remains unknown. On the basis of our structures, FluPol could remain bound to the Pol II surface during the first steps of viral transcription elongation[35] (Extended Data Fig. 9d). However, RNA cleavage probably weakens FluPol binding to Pol II, which seems to correlate with a reduced FluPol occupancy on Pol II in our cryo-EM analysis of the post-cleavage structure. Furthermore, as FluPol transcription progresses, the viral mRNA emerging from the FluPol product exit channel might need more space. These events might lead to FluPol dissociation from the Pol II core, perhaps concomitant with release of the capped RNA from the FluPol cap-binding site and subsequent recruitment of the nuclear cap-binding complex[56]. Alternatively, termination factors such as Integrator or XRN2 may recognize the FluPol-bound Pol II early elongation complex and could compete with FluPol, triggering its dissociation from Pol II surface (Extended Data Fig. 9h,i), although FluPol could remain bound to the Pol II CTD[42]. FluPol dissociation and Pol II recycling would allow another round of cellular transcription, leading to a new 5′ cap that can be snatched again by FluPol.

Our structures of co-transcriptional cap-snatching complexes reveal conserved interfaces between FluPol and transcribing Pol II that are crucial for cap snatching. Targeting and disruption of such small protein–protein interfaces by small-molecule inhibitors is inherently difficult[57]. However, together with recent advances in predicting such interactions[58,59], our results may prompt future in silico and experimental studies to identify suitable compounds.

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

# Methods

## Cloning and purification of proteins

To generate H7N9 FluPol with impaired endonuclease activity, the PA(E119D) mutation was introduced into the PA gene. A pFastBac Dual vector encoding the influenza polymerase heterotrimer subunits of A/Zhejiang/DTID-ZJU01/2013 (H7N9)[35], was used as a template for PCR site-directed mutagenesis and Gibson cloning. This method was also used to generate mutated FluPol variants with altered interface with the Pol II elongation complex. Sequencing of all polymerase subunits confirmed the successful introduction of the site-specific mutations in the PA or PB2 gene.

The wild-type FluPol, FluPol PA(E119D) and other FluPol mutants were essentially expressed and purified as described[35], with the following modification for all experiments, except the sample preparation for the post-cleavage structure. Initial viruses were generated using transfection in Sf9 cells (obtained from ThermoFisher, not verified in-house), virus propagation in Sf21 cells and protein expression in Hi5 cells (both obtained from Expression Systems, not verified in-house). Instead of ammonium sulfate precipitation as a first step during purification, the supernatant was clarified by ultracentrifugation in a Ti45 rotor (Beckman Coulter) at 45,000 rpm and 4 °C for 1 h.

To generate the human transcription factor DSIF with altered binding interfaces with FluPol, SPT5 mutations were introduced. The pETDuet-1 plasmid containing the codon optimized human genes[33] was amplified using site-directed mutagenesis primers, followed by Gibson assembly, transformation and selection. Plasmids were subjected to full plasmid sequencing to check for correct insertions.

The human transcription factors (DSIF and CAK kinase trimer) were expressed and purified as described previously[25,33,60]. Mutated versions of DSIF were not dialysed into 300 mM NaCl buffer, but instead cleaved overnight in the elution buffer and then diluted with no salt buffer to the same salt concentration on the next day. Pol II was purified from pig thymus as described in[33,49], leaving out the size-exclusion step.

## In vitro transcription

The mRNAs were transcribed from two DNA primers[61]. The primers are complementary at the promoter site for the T7 polymerase, and the desired RNA sequence is single-stranded. The in vitro transcription mixture contained 1 μM primers, 40 mM Tris-HCl pH 8.0, 30 mM MgCl$_2$, 2 mM spermidine, 50 mM NaCl, 5 mM NTPs (pH adjusted to 7), 2% DMSO, 0.01% Triton X-100, and 5% T7 DNA-dependent RNA polymerase (homemade). The in vitro transcription reaction was incubated at 37 °C overnight.

The following day, for 1 ml of reaction, 10 μl of Proteinase K (NEB) and 10 μl of DNAse I (ThermoFisher) were added. The reaction was incubated at 37 °C for another 10 min. In addition, 160 μl EDTA (0.5 M pH 8.0) and 80 μl NaCl (5 M) were added to dissolve pyrophosphate precipitates. Then, the RNA was precipitated by adding 900 μl isopropanol and incubating at −80 °C for 2 h. The mixture was centrifuged at 21,000g at 4 °C for 15 min, and the supernatant was discarded. The pellets were air-dried, resuspended in 150 μl RNAse-free water, 2× RNA loading dye was added to 1× (47.5% formamide, 0,01% bromophenol blue, 0.5 mM EDTA) and incubated at 70 °C for 5 min. This mixture was then loaded onto a 12% denaturing urea polyacrylamide gel (8 M urea, 1× TBE (Sigma Aldrich), 12% Bis-Tris acrylamide 19:1 (Carl Roth)) and run in 1× TBE at 300 V for 30 min. Afterward, the gel was covered in plastic wrap and placed on a fluor-coated cellulose TLC plate (Sigma Aldrich) in a darkroom. The RNA bands were visualized using UV shadowing on the TLC plate at 254 nm.

The desired RNA band was cut out from the gel and shredded by passing the gel through two 3 ml syringes. 0.3 M NaOAc pH 5.2 (Invitrogen) was added to cover all gel pieces and incubated at −80 °C overnight. Then, the small pieces were incubated at 37 °C for 30 min and centrifuged at 21,000g for 5 min, and the supernatant was transferred into a fresh tube. This process of adding NaOAc and collecting the supernatant was repeated five times. The supernatants were filtered using a 0.22 μm syringe filter, precipitated with 70% ethanol, and incubated at −80 °C overnight. On the next day, the mixture was centrifuged at 21,000g at 4 °C for 30 min. The pellet was resuspended in RNAse-free water. Then, the RNA was purified using the Monarch RNA Cleanup Kit (500 μg, NEB). The concentration of the RNA was determined by measuring the absorbance at 260 nm using a NanoDrop, microvolume UV/Vis Spectrometer (Thermo Fisher). The RNA was stored at −80 °C until further use.

## Capping of RNAs

The Vaccina capping enzyme system (NEB) was used to generate the 5′ cap structure for the RNAs produced in the in vitro transcription reactions. For the cap(0) structure (m7GpppN-), up to 20 μg uncapped RNA was modified in a 40 μl reaction, containing 1 U μl$^{-1}$ RiboLock (ThermoFisher), 1× capping buffer (NEB), 0.5 mM GTP (ThermoFischer), 0.2 mM S-adenosyl-methionine (SAM, NEB), and 2 μl of Vaccina capping enzyme (homemade, 3 mg ml$^{-1}$). For a cap(1) structure (m7GpppNm-) on the RNA, the reaction described above included another 2 μl of mRNA cap 2′-O-methyltransferase (50 U μl$^{-1}$, NEB). The capping reaction was incubated at 37 °C for 4 h.

Then, the RNA in the reaction was purified using the Monarch RNA Cleanup Kit (50 μg, NEB).

Capping was checked by loading 70 ng of the capped RNAs onto a 20% denaturing urea polyacrylamide gel. The gel was stained with SYBR Gold (1:10,000). The gels were scanned on the Typhoon FLA 9500 (GE Healthcare) for SYBR Gold.

## 3′-Cy5-labelling of RNAs

The RNA was labelled at the 3′ end using RNA liagtion. Up to 5 μg of RNA were used in a 20 μl reaction, containing additionally 0.5 mM ATP (Jena Bioscience), 50 μM Cy5-pCp (Jena Bioscience), 1× buffer (Jena Bioscience), 2 U μl$^{-1}$ RiboLock (ThermoFisher), 1 μL T4 RNA ligase (Jena Bioscience). The mixture was incubated at 16 °C overnight. The labelled RNA was purified using a Monarch RNA Cleanup Kit (10 μg, NEB).

## Endonuclease activity assay

For the endonuclease cleavage assay, 0.05 μM 3′ end Cy5-labelled cap(1)-RNA (m7Gppp2′mrGrArA rGrCrG rArGrA rArGrA rArCrA rCrArGr A rCrArG rCrArG rCrArG rArCrC rArGrG rCr/iCy5C/p) was annealed to 0.05 μM of template DNA (GAT CAA GCT CAA GTA CTT AAG CCT GGT CTA TAC TAG TAC TGC C) in a thermocycler by heating to 72 °C followed by cooling to 4 °C at a rate of 0.1 °C/s. 0.08 μM mammalian Pol II was added to the RNA: DNA hybrid and incubated at 30 °C for 10 min. Then, 0.08 μM non-template DNA (GGC AGT ACT AGT ATT CTA GTA TTG AAA GTA CTT GAG CTT GAT C) was added and incubated at 30 °C for 10 min. Next, 0.12 μM of human elongation factors (DSIF) were added. Furthermore, 0.12 μM CAK and 1 mM ATP were added to generate phosphorylated Pol II. The mixture was incubated at 30 °C for 30 min. After that, 0.01 μM viral WT FluPol or mutated FluPol with equimolar panhandle 5′ vRNA (/5Phos/rArGrU rArGrU rArArC rArArG rArG) and 3′ vRNA (rCrUrC rUrGrC rUrUrC rUrGrC rU) pre-incubated at 4 °C were added. To control for cleavage defect on RNA only, 10× the amount of FluPol and viral RNAs was used. The reactions were incubated at 30 °C, and samples were taken at 0, 10, and 60 min. These reactions occurred in 50 μl with a final buffer composition of 20 mM HEPES pH 7.4, 150 mM NaCl, 4% (v/v) glycerol, 3 mM MgCl$_2$, 1 U μl$^{-1}$ RiboLock (Thermo Fisher), and 1 mM TCEP.

The reactions were stopped by adding 1 μl of Proteinase K (NEB) to 7 μl of the sample and incubation on ice for 5 min. Then, 7 μl of 2× RNA Loading Dye (1× TBE, 3.6 M Urea, 0,01% bromophenol blue) was added to the sample. The samples were loaded onto 20% denaturing urea acrylamide gels and ran in 1× TBE buffer for 75–90 min at 300 V. The gels were scanned at the Typhoon FLA 9500 (GE Healthcare) for Cy5 fluorescence with a sensitivity setting (PTM) of 750.

This protocol was modified in the following way to check for $Mg^{2+}$ dependence of the cap-snatching reaction during the sample preparation for cryo-EM. HEPES pH 7.4 was replaced by Bicine pH 8.5. The $Mg^{2+}$ concentration was altered to 0.1 mM and 3 mM. The ATP concentration was changed to 0.01 mM and 1 mM to avoid complete chelating of $Mg^{2+}$ by ATP. FluPol[E119D] was used instead of wild type. Protein and nucleic acid concentration were change to the concentration used in the sample preparation of cryo-EM in a total volume of 15 µl. The reaction was incubated at 30 °C for 10 min, and then analysed as described above.

## Quantification and statistical analysis of endonuclease assays

The gels of the endonuclease activity assays were quantified using Fiji (v.2.9.0)[62]. Therefore, the lanes were selected using rectangular selection masks. Then, the pixel intensities of each lane were plotted using the built-in gel-analysis functions. The intensity profile from each lane was examined, and individual bands could be distinguished as peaks. Vertical lines were drawn to delimit the peaks. The integrated intensities of each peak were measured and quantified as follows: the intensity of all product bands was divided by the sum of the intensities of all product bands and the substrate band (Extended Data Fig. 1c). The procedure allows us to conclude a normalized cleavage ratio of the FluPol. The results were plotted using Graphpad Prism v.9.4.1, indicating all individual data points as circles.

*P* values were calculated using a two-sided linear mixed-effects model (condition as a fixed effect, experiment as a random effect) with no correction for multiple testing. *P* values are indicated in the figure.

## In vitro FluPol transcription activity assay

For assays, 0.19 µM cap(1)-RNA (rGrArA rGrCrG rArGrA rArGrA rArCrA rCrArGrA rCrArG rCrArG rCrArG rArCrC rArGrG rC) was annealed to 0.19 µM of template DNA in a thermocycler by heating to 72 °C followed by cooling to 4 °C at a rate of 0.1 °C s$^{-1}$. Mammalian Pol II (0.31 µM) was added to the RNA: DNA hybrid and incubated at 30 °C for 10 min. Then, 0.31 µM non-template DNA was added and incubated at 30 °C for 10 min. Next, 0.12 µM of DSIF were added. Furthermore, 0.50 µM CAK and 1 mM ATP were added to generate phosphorylated Pol II. The mixture was incubated at 30 °C for 30 min. After that, 0.62 µM viral FluPol with modified panhandle vRNAs (3′ vRNA with high G content, rCrUrG rUrGrU rGrCrC rUrCrU rGrCrU rUrCrU rGrCrU and 5′ vRNA /5Phos/rArGrU rArGrU rArArC rArArG rArG) pre-incubated at 4 °C were added. Furthermore, 0.10 µM of CTP and GTP were added, as well as 0.77 µCi µl$^{-1}$ α-$^{32}$P-CTP. The reactions were incubated at 30 °C for 2 h. These reactions occurred in 12.9 µl with a final buffer composition of 20 mM HEPES pH 7.4, 150 mM NaCl, 4% (v/v) glycerol, 3 mM $MgCl_2$, 1 U µl$^{-1}$ RiboLock (Thermo Fisher), and 1 mM TCEP.

The reactions were stopped by adding 1 µl of Proteinase K (NEB) to the sample and incubation at 37 °C for 15 min. Then, 14 µl of 2× RNA Loading Dye (1× TBE, 3.6 M Urea, 0,01% bromophenol blue) was added to the sample. The samples were loaded onto 20% denaturing urea acrylamide gels and ran in 1× TBE buffer for 75 min at 300 V. The gels were incubated for 2 h on a phosphorus screen. The screen was scanned at the Typhoon FLA 9500 (GE Healthcare) with PTM = 800.

## Analytical gel filtration on Äkta µ

For an assembly in a 50 µl reaction, 42.75 pmol RNA was annealed to 42.75 pmol template DNA as described for the endonuclease assay. 28.5 pmol mammalian Pol II was added to the RNA:DNA scaffold, followed by 57 pmol of non-template DNA, and incubated at 30 °C for 10 min after each addition. Next, 0.8 µM CAK, 1 mM ATP, and 57 pmol human transcription elongation factors were added and incubated at 30 °C for 30 min. The CAK was omitted for the non-phosphorylation assays. Then, pre-mixed 57 pmol viral FluPol (endonuclease inactive version PA(E119D)) with equimolar panhandle 5′ vRNA (/5Phos/rArGrU rArGrU rArArC rArArG rArG) and 3′ vRNA (rCrUrC rUrGrC rUrUrC

rUrGrC rU) were added to the mix. Last, the reaction was incubated at 30 °C for an additional 10 min. The final buffer composition was 50 mM Bicine pH 8.5 at 4 °C, 150 mM NaCl, 4% (v/v) glycerol, 3 mM $MgCl_2$ and 1 mM TCEP.

The fully formed complex was centrifuged at 21,000g at 4 °C for 10 min. The supernatant was injected onto a Superose 6 Increase 3.2/300 column (Cytiva) and ran in SEC buffer (20 mM Bicine pH 8.5 at 4 °C, 150 mM NaCl, 4% (v/v) glycerol, 3 mM $MgCl_2$, 1 mM TCEP) on an ÄKTAmicro (GE Healthcare) system. The absorbances at 280 nm (protein) and 260 nm (RNA/DNA) were measured. The absorbance data were plotted using GraphPad Prism v.9.4.1. The main elution fractions were analysed by SDS–PAGE.

## Western blot

Samples of the peak fractions were collected to compare the presence of FluPol in the Pol II containing fractions, mixed with 4× SDS-loading dye (ThermoFisher), and stored at −20 °C until analysis.

The samples were run on one SDS–PAGE (NuPAGE 4–12% Bis-Tris, Invitrogen) in 1× MES buffer (Invitrogen). The gel was then blotted onto a nitrocellulose membrane (GE Healthcare) using a wet-blot system (ThermoFisher) in NuPAGE transfer buffer (Invitrogen). The blot was then blocked for 1 h at room temperature with 5% (w/v) milk powder in PBS-T. Then, the membrane was cut horizontally at the 50 kDa line. The upper half was incubated overnight with a rabbit anti-Strep antibody (1:1,000 dilution; ab76949, Abcam) against the StrepTag II on the FluPol. The lower half was incubated with a rabbit anti-RPB3 polyclonal (1:2000 dilution; A303-771A, Bethyl) as a loading control.

The following day, the membranes were washed 3 × 1 min and 3 × 10 min with PBS-T and incubated with an anti-rabbit antibody coupled to horseradish peroxidase (1:1,000; NA937, GE Healthcare) in PBS-T with 5% milk powder. Then, the membrane was washed three times with PBS-T for 10 min, developed with SuperSignal West Pico Substrate (Thermo Fisher), and scanned using a ChemoCam Advanced Fluorescence imaging system (Intas Science Imaging).

To assess steady-state levels of A/WSN/33-derived PA and PB2 proteins, total lysates of HEK-293T cells transfected with the corresponding pcDNA3.1 expression plasmid were prepared in Laemmli buffer. Proteins were separated by SDS–PAGE using NuPAGE™ 4–12% Bis-Tris gels (Invitrogen) and transferred to nitrocellulose membranes which were incubated with primary antibodies directed against PA (GTX125932, 1:5,000), PB2 (GTX125925, 1:5,000) or tubulin (Sigma Aldrich T5168, 1:10,000) and subsequently with horseradish peroxidase-tagged secondary antibodies (Sigma Aldrich, A9044 and A9169, 1:10,000). Membranes were developed with the ECL2 substrate according to the manufacturer's instructions (Pierce) and chemiluminescence signals were acquired using the ChemiDoc imaging system (Bio-Rad). Uncropped gels are provided as a source data file.

## Sample preparation for Cryo-EM

First, 180 pmol cap(1)-RNA was annealed to 180 pmol 5′-Cy5-labelled template DNA, as stated previously. 120 pmol mammalian Pol II was added to the RNA-DNA scaffold and incubated at 30 °C for 10 min. Then, 240 pmol of non-template was added and kept at 30 °C for 10 min. Next, 1 µM CAK, 1 mM ATP and 240 pmol human transcription elongation factors were added and incubated at 30 °C for 30 min. Last, pre-mixed 240 pmol viral FluPol (endonuclease inactive version PA(E119D)) with equimolar 5′ and 3′-vRNAs was added to the mix and incubated at 30 °C for 10 min. The 3′-vRNA was ATTO532-labelled on the 5′ end. The complex was assembled in a buffer containing 50 mM Bicine pH 8.5 at 4 °C, 150 mM NaCl, 4% (v/v) glycerol, 0.1 mM $MgCl_2$ (3 mM $MgCl_2$ for post-cleavage conformation, 0.1 mM $MgCl_2$ for pre-cleavage conformation), and 1 mM TCEP in a volume of 150 µl. The fully formed complex was centrifuged at 21,000g at 4 °C for 10 min.

The sample was loaded on a continuous 10–40% glycerol gradient containing assembly buffer components. The heavy solution contained

additionally 0.1% (v/v) glutaraldehyde. The gradient was centrifuged at 33,000 rpm in a SW60 rotor (Beckman Coulter) at 4 °C for 16 h. The next day, the gradient was fractionated in 200 µl fractions. The cross-linker was quenched by adding 100 mM Tris-HCl pH 8.0 at 4 °C. Fractions were analysed by NativePAGE 3–12% (Bis-Tris, Invitrogen) run at 4 °C. The gel was then scanned for Cy5 and ATTO532 signals, followed by Coomassie staining.

Then, the complex containing fractions were dialysed against 20 mM Tris pH 8 at 20 °C, 20 mM Bicine pH 8.5 at 4 °C, 100 mM NaCl, 4% (v/v) glycerol, 0.1 mM $MgCl_2$ (3 mM $MgCl_2$ for post-cleavage conformation, 0.1 mM $MgCl_2$ for pre-cleavage conformation), and 1 mM TCEP using a 20 kDa Slide-A-Lyzer MINI device (Thermo Fisher) at 4 °C for 4 h. Onto the sample was a continuous carbon film of roughly 3 nm floated for 5 min. The carbon was then fished with a glow-discharged holey carbon grid (Quantifoil R3.5/1, copper, mesh 200). Four microlitres of dialysis buffer was added to the grid, and the grid was placed in a Vitrobot Mark IV (Thermo Fisher) under 100% humidity at 4 °C. The grids were then blotted using Whatman paper with a blot force of 5 for 5 s and directly plunge-frozen in liquid ethane.

### Cryo-EM analysis and image processing
A Titan Krios G2 transmission electron microscope (FEI) operated at 300 keV, equipped with a GIF BioQuantum energy filter (Gatan) and a K3 summit direct detector was used to acquire cryo-EM data. Data acquisition was performed at a pixel size of 1.05 Å per pixel using Serial EM, corresponding to a nominal magnification of 81,000× in nanoprobe EFTEM mode.

The pre-cleavage dataset was collected in 5 batches. A total of 60,032 movie stacks were collected. Each movie contained 40 frames and was acquired in counting mode over 1.95 s. The defocus was set to values between −0.1 to −2.0 µm. The dose rate was 20.48 $e^-$ $Å^{-2}$ $s^{-1}$, leading to a total dose of 39.94 $e^-$ $Å^{-2}$.

The post-cleavage dataset was collected in 3 batches. A total of 20,509 movie stacks were collected. Each movie contained 40 frames and was acquired in counting mode over 2.4 s. The defocus was set to values between −0.1 to −2.0 µm. The dose rate was 18.34 $e^-$ $Å^2$ $s^{-1}$, leading to a total dose of 40 $e^-$ $Å^{-2}$.

Data preprocessing, including stacking, contrast transfer function (CTF) estimation, and dose-weighting, was done using Warp[63]. In Warp, particles were also picked using an on this data set trained version of the neural network BoxNet2.

For the post-cleavage dataset, 11,935,228 particles were extracted in five batches in RELION-3.1.0 (ref. 64) using a binning factor of four. The box size of the particles was set to 112 pixels with a pixel size of 4.2 Å per pixel. The particles were then imported into cryoSPARC (v.4.3.1)[65]. In cryoSPARC, particles that do not align were removed, as well as particles that do not contain Pol II using 3D heterogeneous refinements. The 1,975,313 particles that contain Pol II were transferred to RELION and extracted with a box size of 448 pixels and a pixel size of 1.05 Å pixels. The particles were refined using a mask around the Pol II core, followed by Bayesian polishing and CTF refinement for beam tilt and per-particle defocus values. The particles were reloaded into cryoSPARC, combined into three datasets, followed by one round of heterogeneous refining for FluPol occupancy. Then, the datasets were individually non-uniformly refined, locally refined onto the FluPol, and 3D classified. The data sets were merged, locally refined for FluPol, and two times 3D classified. From a final dataset of 63,230 particles, focus refinements on FluPol, Pol II core, Pol II stalk and the interface were performed.

For the pre-cleavage dataset, 6,423,874 particles were extracted in three batches in RELION-3.1 (ref. 64) using a pixel size of 4.2 Å per pixel and a box size of 112 pixels. The particles were then imported into cryoSPARC (v.4.3.1)[65]. In cryoSPARC, particles that do not align were removed, as well as particles that do not contain Pol II using 3D heterogeneous refinements. The 1,937,625 particles that contain Pol II were transferred to RELION and extracted with a box size of 448 px and a pixel size of 1.05 Å/px. These particles were refined using a mask around the Pol II core, followed by Bayesian polishing and CTF refinement for beam tilt and per-particle defocus values. The particles were focused refined, and classified on the Pol II core, taking only the particles of the class with good-looking Pol II. These particles were globally classified for FluPol occupancy and then focussed classified on FluPol for well-aligning FluPol particles. This final particle set of 369,858 particles was focused refined on Pol II, CTF refined, and Bayesian polished. Focus refinements for Pol II core and FluPol were performed on the basis of the obtained consensus refinement.

### Model building
For both structures, initial models of *S. scrofa domesticus* Pol II (PDB: 7B0Y[66]), SPT5 KOW2, KOW3, KOWx-4 and KOW5 domains (PDB: 5OIK and 5OHO[33]) and FluPolA/H7N9 (PDB:7QTL[35]) were rigid body fitted in ChimeraX 1.6.1 (ref. 67) using the consensus refinement. The RNA and DNA were manually adjusted in Coot[68] to fit the sequences used in this study. As the density of the RNA in the endonuclease site is not well enough resolved to call a sequence, we modelled the sequence according to the biochemistry. The linker between KOWx-4 and KOW5 was manually built as well, assuming that the best visible amino acid at the G1 nucleotide is the first phenylalanine of the linker. This model, the focused maps, and the consensus map were loaded into ISOLDE 1.6.0 (ref. 69). The focused maps were aligned to the consensus map in ChimeraX. In ISOLDE, Molecular Dynamic simulation was performed using the starting model restrains. Then, the individual protein components were subjected to Real Space Refinement in PHENIX[70] and manual curation in Coot.

For the pre-cleavage conformation, Pol II and KOW5 were refined against the focused map for Pol II. FluPol was refined against the focused map for the FluPol. KOWx-4 was refined against the consensus map.

Pol II (except RPB4 and RPB7) was refined against the Pol II-focused map for the post-cleavage state. SPT5 KOW2, KOW3, KOWx-4, RPB4 and RPB7 were refined against the stalk-focused map. FluPol was refined against the focused map for the FluPol.

Then, the Pol II elongation complex components and the FluPol were rigid body docked into the consensus map in ChimeraX before manually checking interface residues in Coot using the consensus map. The density for SPT4 and SPT4 NGN and KOW1 domain is not well resolved, so the consensus map was lowpass filtered to 6 Å. A deposited model (PDB: 5OIK for pre-cleavage, and 7YCX[71] for post-cleavage) for these domains was rigid body fitted into this filtered map using ChimeraX. Interfaces were checked for major clashes. Clashing residues without density were modified in Coot using the most likely non-clashing rotamer.

To determine the range of possible RNA lengths, a series of FluPol structures was modelled in Coot by cropping nucleotides in the less-resolved space between cap-binding domain and endonuclease domain. Then, these structures were loaded into ISOLDE and the RNA was real-space refined. The lower limit was defined as when ISOLDE shifted the RNA through the endonuclease domain. The upper limit was determined by incrementally increasing the RNA length in Coot, refined in ISOLDE, and visually inspected until the obtained model deviated from an expected linear RNA geometry.

### Selection of interface residues for mutational analysis
First, a list of 38 amino residues at the interfaces was generated, see Supplementary Table 2. In SnapGene, 7.0.1 two MUSCLE alignments were performed for PA and PB2 (Extended Data Fig. 4a,b). Each alignment contained sequences of 6 influenza A viruses, 2 influenza B viruses, and one influenza C and D virus. Sequences of the following strains were used: A/Zhejiang/HZ1/2013 (H7N9), A/WSN/1933 (H1N1), A/California/04/2009 (H1N1), A/California/04/2009 (H1N1), A/Victoria/3/1975 (H3N2), A/Little-yellow-shouldered-bat/Guatemala/2010 (H17N10),

B/Lee/1940, B/Memphis/13/2003, C/Johannesburg/1/1966, D/Bovine/Minnesota/628/2013. The 16 residues with a MUSCLE score of above 50 were considered conserved. These amino acids were mutated to alanine. All mutants were checked for the expression level. The mutant Y131A showed a reduced expression level and was consequently excluded for further analysis. We tested furthermore the following double and triple mutants: PA S140A and E141A; PB2 R375A and R380A; and PB2 D466A, T468A and S470A. From these mutants, only PB2 R375A and R380A had wild-type expression levels.

To investigate the evolutionary conservation of the binding interfaces between mammals and birds, four mammalian species, four bird species and *Caenorhabditis elegans* were used. Sequences were identified using the BLAST algorithm of Uniprot using the selected species as a search target. To select for the bird species, the human RPB1 sequence was blasted against all avian protein sequences available in Uniprot. Only four bird species had full-length annotated RPB1. These species were used as a search filter while blasting human RPB3, RPB11 and SPT5. A list of all Uniprot sequence IDs is available upon request. The obtained sequences were aligned in Snapgene using the ClustalOmega algorithm. Alignments are depicted in Extended Data Fig. 4d–g.

## Cell-based minigenome assay

The plasmids and procedure used for minigenome assays are described in ref. 42. The primers used for mutagenesis of the PB2 and PA plasmids can be provided upon request. In brief, $3 \times 10^4$ HEK-293T (obtained from ATCC, authenticated by ATCC using STR profiling, tested for mycoplasma) cells were co-transfected with plasmids encoding the vRNP protein components (PB2, PB1, PA, NP) from the A/WSN/33 (WSN) virus, a pPoII-Firefly plasmid encoding a negative-sense viral-like RNA expressing the firefly luciferase and the pTK-Renilla plasmid (Promega) as an internal control. Mean relative light units (RLUs) produced by the firefly and Renilla luciferase, reflecting the viral polymerase activity and transfection efficiency, respectively, were measured using the Dual-Glo Luciferase Assay System (Promega) on a Centro XS LB960 microplate luminometer (Berthold Technologies, MikroWin v.4.41) at 24 h post-transfection (hpt). Firefly luciferase signals were normalized with respect to Renilla luciferase. To quantify steady-state levels of mRNA, complementary RNA (cRNA) and vRNA, $3 \times 10^5$ HEK-293T cells were seeded in 12-well plates and transfected with plasmids encoding the vRNP components (PB2, PB1, PA and NP), along with 5 ng per well of a WSN-NA RNA-expressing plasmid. Total RNA was extracted 24 h after transfection using RNeasy Mini columns (Qiagen), following the manufacturer's protocol. Strand-specific RT–qPCR was then performed[46]. In brief, reverse transcription was carried out using primers specific to the viral NA mRNA, cRNA, vRNA and the cellular GAPDH mRNA, with the SuperScript III Reverse Transcriptase (Invitrogen). Quantification was done using SYBR Green (Roche) on the LightCycler 480 system (Roche, Software v.1.5.0.39). RNA levels were normalized to GAPDH when indicated, and relative expression was calculated using the $2^{-\Delta\Delta CT}$ method[72].

## Production and characterization of recombinant viruses

The recombinant WSN virus mutants were produced by reverse genetics. In brief, a mix of $3 \times 10^5$ MDCK (obtained from National Influenza Center, not authentified in-house, tested for mycoplasma) and $4 \times 10^5$ HEK-293T cells were plated in 6-well plate one day before transfection with a mix of 4 expression plasmids for WSN-PB1, PB2, PA and NP proteins and 8 PolI-based plasmids for the 8 viral RNAs, one of the latter carrying a mutation (PA(Y131A), PA(K104A/E141A), PB2(E452R), PB2(D466R), PB2(E452A/D466A) or PB2(K482E)) in Opti-MEM (Gibco) using FuGene6 (Promega) according to the manufacturer's instructions. The following day, cells were washed twice with DMEM and incubated for 48 h at 37 °C in DMEM containing TPCK-treated Trypsin (Sigma, 0.5 µg ml⁻¹). Viral reverse genetic supernatants were collected, centrifuged and titrated on MDCK cells by plaque assay[73]. For viral

amplification, MDCK cells were infected with the reverse genetic supernatants at an MOI of 0.001 or 0.0001 and incubated for 72 h at 37 °C in DMEM containing TPCK-treated Trypsin (Sigma, 1 µg ml⁻¹). The viral supernatants (P1) were titrated by plaque assay and plaque diameters were measured using the Fiji software. Viral RNA was isolated from P1 viral stocks using the QIAamp Viral RNA Mini kit (Qiagen). The eight genomic segments were subjected to reverse transcription and amplification[74]. Next generation sequencing was performed using the Nextera XT DNA Library Preparation kit (Illumina), the NextSeq 500 sequencing systems (Illumina) and the IGV 2.19_4 software for analysis.

## Reporting summary

Further information on research design is available in the Nature Portfolio Reporting Summary linked to this article.

## Data availability

The electron density reconstructions and final models were deposited with the Electron Microscopy Data Bank (accession codes EMD-50892 and EMD-50927) and the PDB (accession codes 9FYX and 9G0A). For structural modelling and comparisons, we used PDB models 4WSB, 5M3H, 5OHO, 5OIK, 6GMH, 6GML, 6RR7, 7B0Y, 7QTL, 7YCX, 8JCH, 8PNP, 8P4F, 8RBX and 8R3L. The next generation sequencing raw reads generated in this study have been deposited in the European nucleotide archive under accession code ERP181209 (https://www.ebi.ac.uk/ena/browser/view/PRJEB98852). Source data are provided with this paper.

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

**Acknowledgements** We thank R. Muir and N. Iwicki for help with purifying FluPol variants; F. Grabbe for providing purified CAK; U. Steuerwald for support at the electron microscope and maintaining the cryo-EM facility; T. Schulz for providing pig thymus tissue; P. Rus and U. Neef for running the insect cell facility; J. Walshe and M. Ochmann for advice on data processing; S. Vos, C. Bernecky and T. Kouba for help with preliminary experiments; S. Paisant for help with plasmid mutagenesis; and V. Enouf and J. Bourret for next generation sequencing analysis.

**Author contributions** A.H.R., M. Lidschreiber, M. Lukarska, C.D., P.C., N.N. and S.C. designed the study. A.H.R. and D.L. planned all experiments except for the cell-based minigenome assays. M.D., T.K. and N.N. planned cell-based assays. A.H.R., D.L., C.O. and A.S. prepared protein components. A.H.R., D.L. and U.N. performed biochemical assays and prepared samples for cryo-EM. A.H.R., I.F. and C.D. acquired and analyzed cryo-EM data. A.H.R. and S.C. built the molecular models. M. Lukarska pioneered reconstitution of the cap-snatching

complex[30]. T.K. and M.D. performed cell-based assays. D.L. and A.H.R. designed figures. A.H.R. and C.D. wrote the manuscript with input from all authors. All authors read and approved the final manuscript. Conceptualization: P.C. and S.C.; methodology: A.H.R., D.L. and S.C.; formal analysis: A.H.R., D.L., I.F., M. Lidschreiber, M.D., T.K. and N.N.; investigation: A.H.R., D.L., M.D., U.N., T.K. and M. Lukarska; writing (original draft preparation): A.H.R.; visualization: D.L. and A.H.R.; writing (review and editing): A.H.R., D.L., M.D., T.K., M. Lidschreiber, M. Lukarska, C.D., P.C., N.N. and S.C; funding acquisition: P.C., A.H.R. and N.N.; resources: A.H.R., D.L., C.O. and A.S.; supervision: P.C., M. Lidschreiber, C.D., S.C. and N.N.

**Funding** Open access funding provided by Max Planck Society.

**Competing interests** The authors declare no competing interests.

**Additional information**
**Correspondence and requests for materials** should be addressed to Nadia Naffakh, Christian Dienemann, Stephen Cusack or Patrick Cramer.

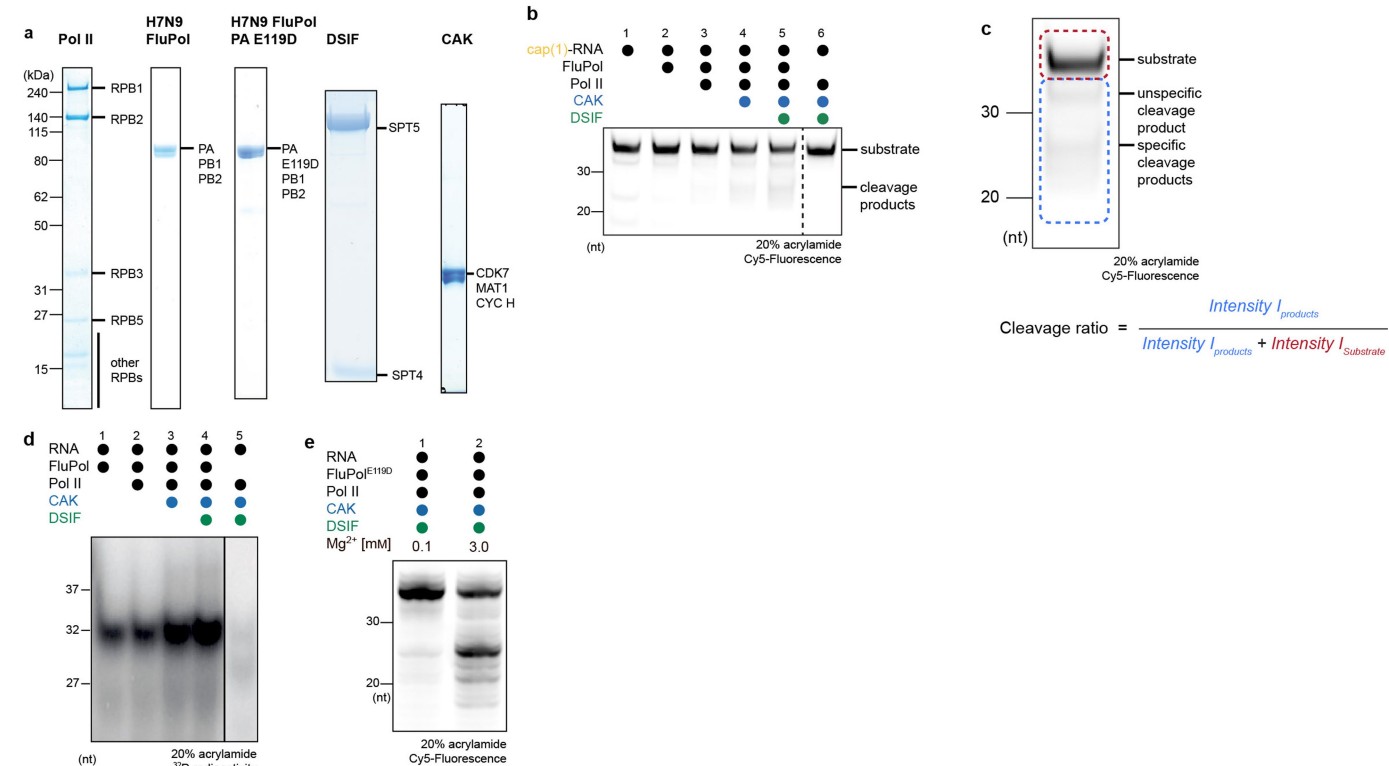

**Extended Data Fig. 1 | Related to Fig. 1. FluPol recognizes the Pol II EC.**
**a**, Representative images of Coomassie-stained SDS-PAGE lanes of the purification of the single protein components used (n = 7 for Pol II, n = 2 for both FluPol proteins, N = 3 for DSIF and CAK). Protein band assignments are based on size. **b**, Representative example for the denaturing urea PAGE of an endonuclease assay using FluPol^WT (n = 6). The substrate and the product bands

are labeled. **c**, Schematic of the in vitro cleavage ratio determination **d**, Phosphoimage of denaturing urea PAGE radioactivity elongation assay showing increased FluPol transcription upon DSIF addition and CTD phosphorylation using FluPol^WT (n = 2). **e**, Denaturing urea PAGE of an endonuclease assay with varying Mg^2+ concentrations using FluPol^E119D (n = 2).

**a**

GraFix: 10-40% Glycerol Gradient w/ 0.1% glutaldehyde
35nt cap(1)-RNA , Pol II, FluPol_E119D, DSIF, CAK, ATP

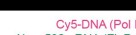

Fractions 10  11  12  13  14  15  16  17  18  19  20

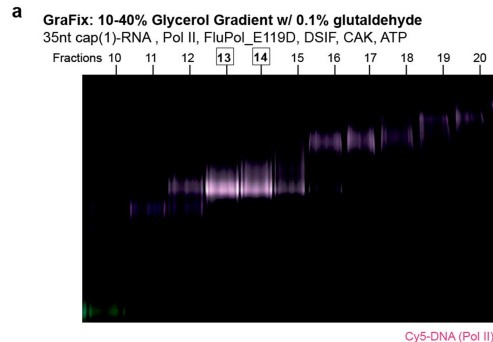

Cy5-DNA (Pol II)
Alexa532-vRNA (FluPol)

**b**

20,509 movies
motion correction and CTF estimation in Warp

denoised micrograph

6,423,874 particles picked in Warp
particle extraction in RELION 4.2 Å/px
112 x 112 px
Extensive cleanup in cryoSPARC in

1,937,625 particles
Non-uniform refinement

1,937,625 particles
particle extraction in RELION 1.05 Å/px
448 x 448 px
combine subsets

3D refinement
Pol II mask in transparent and cyan

foc. refinement on Pol II
CTF refinement
Bayesian polishing
3D classification on Pol II

57%   24%

19%   0.2%

631,345 particles
foc. refinement FluPol
3D class on FluPol

~90°

29%   23%

24%   25%

879,282 particles
Pol II foc. refinement
global classification

15%   45%   5%

16%   8%   10%

369,858 particles
Pol II foc. refinement
CTF refinement
Bayesian polishing
Overall refinement

**c**

Consensus map ~3.3Å

**d**

~90°

FluPol Mask

Focussed refinements

~90°

Pol II mask

**e**

~90°

FluPol map ~2.9 Å

Pol II map ~2.9 Å

**f**

FSC Resolution estimate

Consensus map
FSC
Resolution 1/Å
3.29 Å

FluPol map
FSC
Resolution 1/Å
2.90 Å

Pol II map
FSC
Resolution 1/Å
2.94 Å

**g**

Local resolution

Consensus map

FluPol map

2.8  3.3  3.8  4.3  6.0

~90°

Pol II map

**h**

Angular distribution

Consensus map
7218  4812  2406  1

FluPol map
7668  5112  2556  1

Pol II map
6864  4576  2288  1

**Extended Data Fig. 2 | Data acquisition and processing of the pre-cleavage complex. a**, Fluorescence scan of the native PAGEs analyzing GraFix gradient fractions of the pre-cleavage complex preparation (n = 1). The image is an overlay of Cy5 and ATTO532 signals. In magenta is the scan for the Cy5 channel of the labeled template DNA, and in green is the scan for the ATTO532 channel of the labeled 3′ vRNA. Highlighted fractions were combined and used for cryo-EM analysis. **b**, Flowchart illustrating the key steps of the processing pipeline for obtaining the structure of the pre-cleavage complex. **c**, Consensus density map of the pre-cleavage FluPol-Pol II-DSIF EC complex colored by underlying protein components. **d**,**e**, Masks used for focused refinements and their resulting maps. **f**, Gold-Standard Fourier shell correlation plots of the consensus and focused maps. **g**, Local resolution of consensus and focus refinement maps. Local resolution estimations were performed in RELION. **h**, Angular distribution of the consensus refinement and focused refinements plotted with Warp.

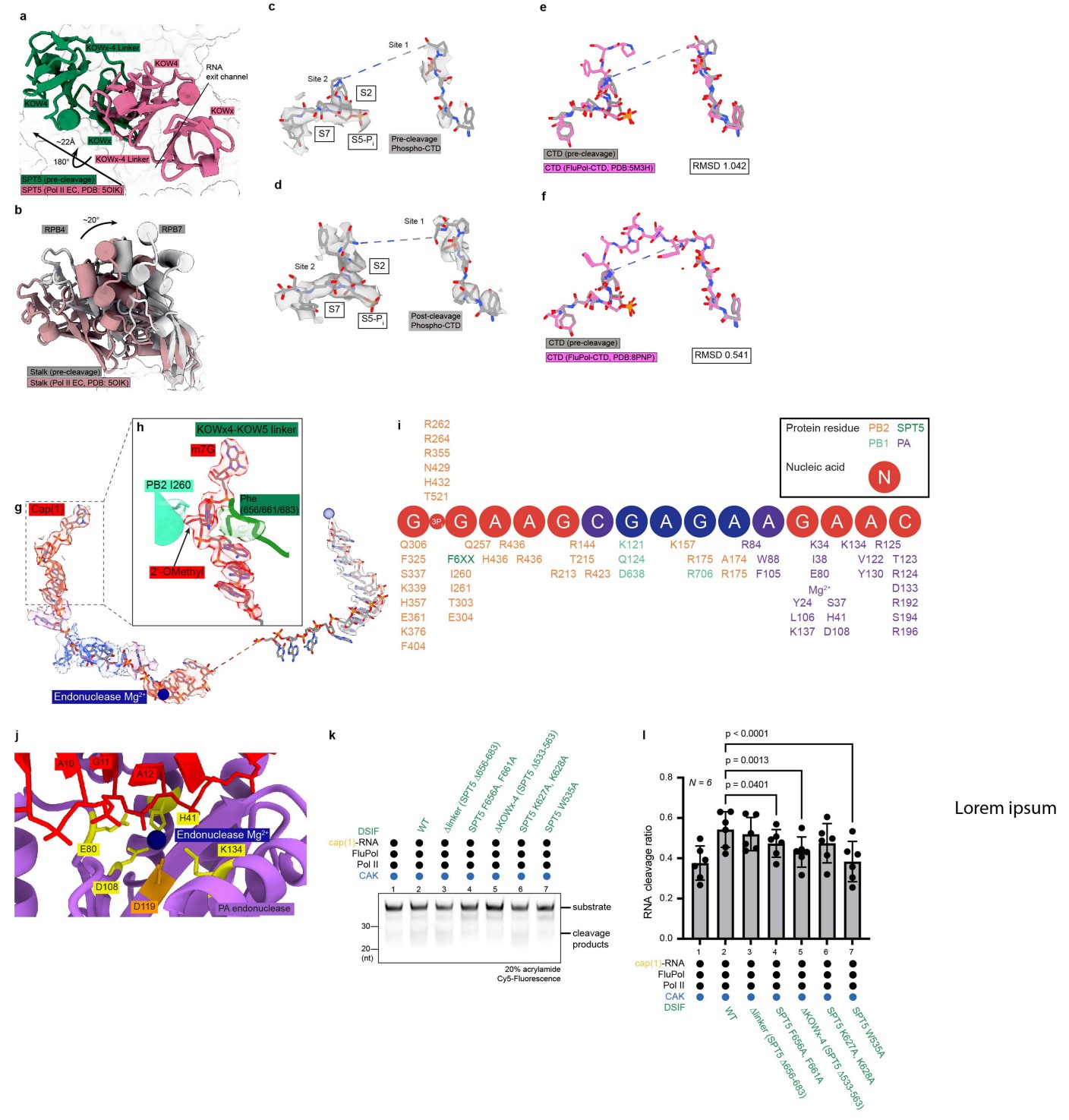

Lorem ipsum

**Extended Data Fig. 3 |** See next page for caption.

**Extended Data Fig. 3 | Related to Fig. 2. Structure of the pre-cleavage cap-snatching complex. a-b**, Comparison of pre-cleavage structure with the canonical Pol II-DSIF EC (PDB:5OIK)[33]. **a**, The canonical position of the KOWx-4 domain is shown in pink, and the position observed in the pre-cleavage structure is shown in green. The KOWx-4 domain of DSIF SPT5 is displaced by -22 Å and rotated by -180° relative to the canonical conformation. **b**, The canonical position of the Pol II stalk is colored in dusty pink, and the stalk model from the pre-cleavage structure is depicted in gray. The Pol II stalk is moved by 20° around its base relative to the canonical conformation. **c**, Phosphorylated CTD of RPB1 bound to FluPol shown in sticks. The obtained cryo-EM density of the pre-cleavage structure is displayed in transparent gray. **d**, Phosphorylated CTD of RPB1 bound to FluPol shown in sticks. The obtained cryo-EM density of the post-cleavage structure is displayed in transparent gray. **e,f**, Comparison of CTD binding the prior structures[2,42] shows a similar conformation of the CTD in the CTD-binding site of FluPol. **g**, Cryo-EM density for the RNA within FluPol between the PB2 cap-binding and PA endonuclease domains. The density is colored by thresholds. Red corresponds 0.015, purple to 0.0075 and blue to 0.004. The endonuclease active site $Mg^{2+}$ and Pol II polymerase active are shown in dark blue spheres. RNA within Pol II is shown and the corresponding density is shown in transparent gray. The density was restricted to areas close to the RNA in ChimeraX. The RNA is colored by heteroatom. **h**, Zoom in to 5′ mRNA cap(1) inside the PB2 cap-binding domain and the supporting Phenylalanine from the KOWx-4-KOW5 linker. The supporting Isoleucine260 of PB2 interacting with the 2′-Methoxygroup is highlighted as well. The density for the RNA and the linker is transparent and colored in the color of the underlying models. **i**, Schematic representation of the RNA and its interacting residues within the complex. The RNA is color coded by resolution using the same thresholds as in **g**. **j**, Active center of the endonuclease domain is shown. Magnesium chelating residues are show in yellow. In orange is the mutated PA E119D residue used in the structure. **k,l** Effect of mutation of DSIF on the stimulatory effect of DSIF on the endonuclease activity of FluPol in vitro. Representative example for the denaturing urea PAGE of an endonuclease assay is shown in **k**. The substrate and the product bands are labeled. Fraction of cleaved RNA (intensity of cleaved product divided by intensity of all bands, see Extended Data Fig. 1c) in dependence of factors added. Each point reflects one experimental replicate (n = 6), shown as mean ± s.d. Significance p-values were calculated using a linear mixed-effects model (substrate as a fixed effect, experimental replicate as a random effect, two-sided, no multiple testing correction). Comparison of DSIF WT and DSIF SPT5 F656A, F661A; DSIF SPT5 Δ552-563; DSIF SPT5 W535A have p-values of 0.0401; 0.0013 and <0.0001.

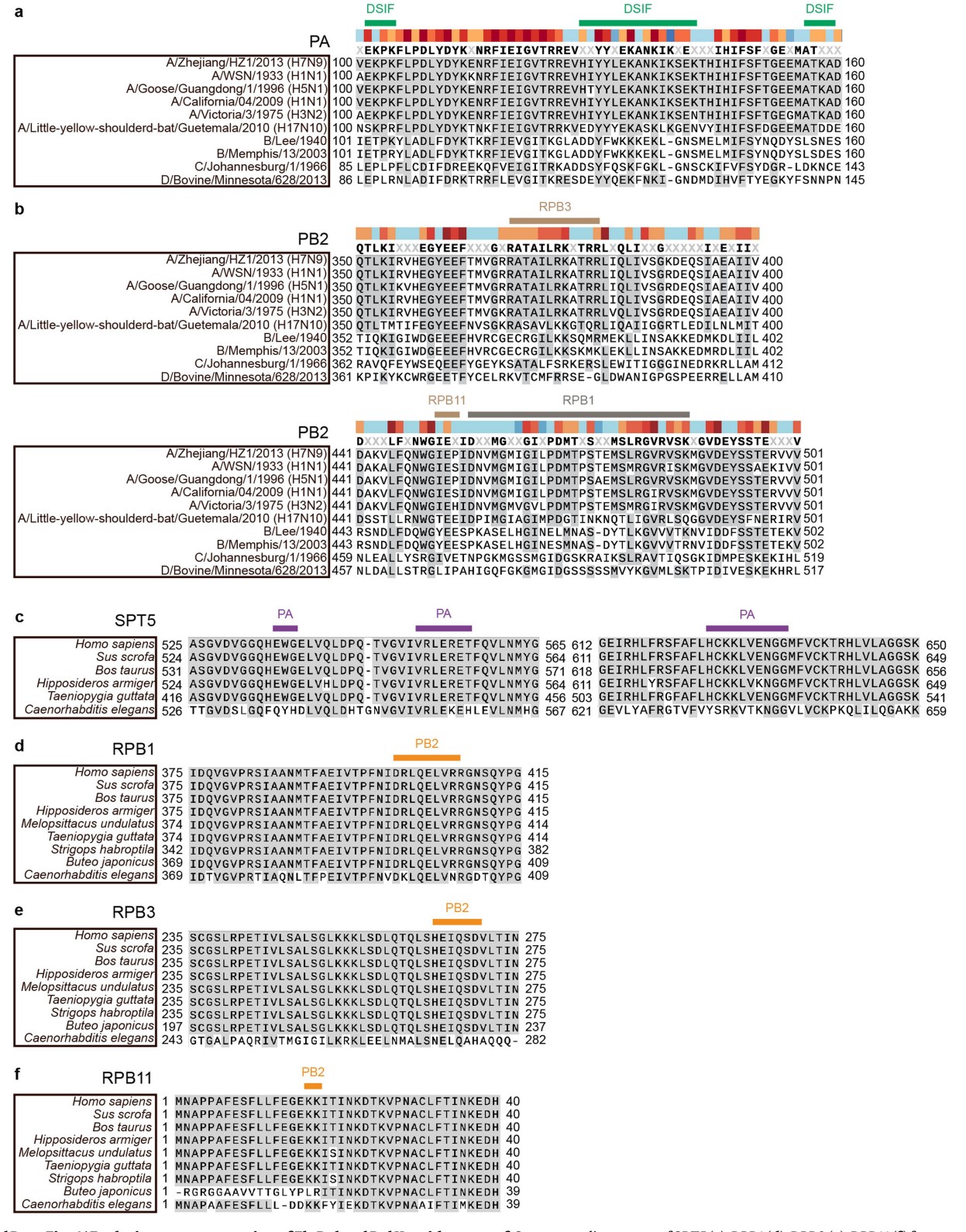

**Extended Data Fig. 4 | Evolutionary conservation of FluPol and Pol II residues involved in the interface. a**, Sequence alignment of PA from representative influenza A–D strains. Residue stretches interacting with DSIF are highlighted with a green box above the alignment. **b**, Sequence alignment of PB2 from representative influenza A–D strains. Residue stretches interacting with different RPBs are highlighted with boxes in different colors above the alignment. **c-f**, Sequence alignments of SPT5 (**c**), RPB1 (**d**), RPB3 (**e**), RPB11 (**f**) from mammals (*H. sapiens, S. scrofa, B. taurus, H. armiger*), birds (*M. undulatus, T. guttata, S. habroptila, B. japonicus*) and *C. elegans*. Residue stretches interacting with FluPol are highlighted with a box above the alignment. Bird species selection was based on well annotated RPB1. SPT5 was not well annotated in *M. undulatus, S. habroptila* and *B. japonicus*, consequently, they were omitted in panel **e**.

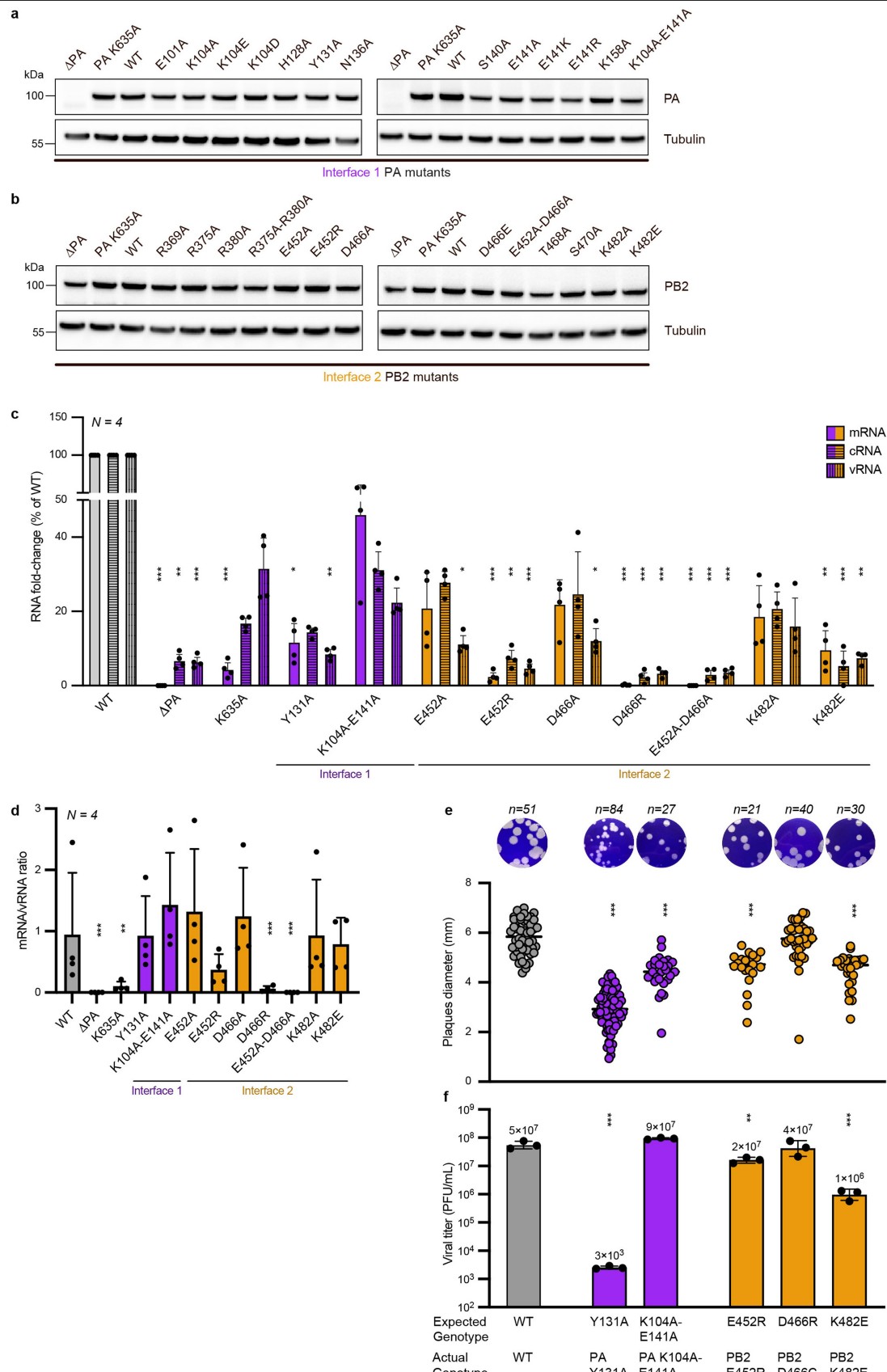

**Extended Data Fig. 5** | See next page for caption.

**Extended Data Fig. 5 | In vivo effects of FluPol mutations involved in the interface to the Pol II EC. a**,**b**, Western blots against PA (**a**), PB2 (**b**), and tubulin (**a**,**b**) for wild type (WT) and mutant FluPol variants transiently expressed in HEK-293T cells. **c**, The Steady-state levels of NA mRNA, cRNA and vRNA in a minigenome assay setting are measured by ss-RTqPCR. The WT or indicated mutant WSN vRNPs were reconstituted in HEK-293T cells in the presence of the NA vRNA. The PA-K635A mutant was used as a transcription defective control[2]. Total RNAs were analyzed by strand-specific RTqPCR[46] **c**, The data are represented as percentages (100%: WT) and as the mean ± s.d. Each point reflects one biological replicate (n = 4). *p < 0.05, **p < 0.01, ***p < 0.001, 2-way ANOVA on log-transformed data with Tukey's multiple comparison test (black stars) or Dunnett's multiple comparison test (colored stars) using the WT as a reference. **d**, Ratio of mRNA/vRNA levels based on measurements in **c** shown as the mean ± s.d. (N = 4). *p < 0.05, **p < 0.01, ***p < 0.001, one-way ANOVA on log-transformed data with Dunnett's multiple comparison test using the WT as a reference. **e**,**f**, Mutant viruses rescued by reverse genetics were characterized phenotypically by plaque measurement (**e**) and titration (**f**) and genotypically by full genome sequencing using Illumina sequencing (expected and actual genotypes are indicated). Each dot represents one plaque (**e**) or one titration (**f**), virus titer is depicted as mean ± s.d. (n = 3 technical replicates). *p < 0.05, **p < 0.01, ***p < 0.001, one-way ANOVA one-way ANOVA on log-transformed data with Dunnett's multiple comparison test using the WT as a reference. Actual genotypes summarize mutations that occurred in the indicated segment and in more than 50% of the sequencing reads relative to WT. For detailed description see Table 1.

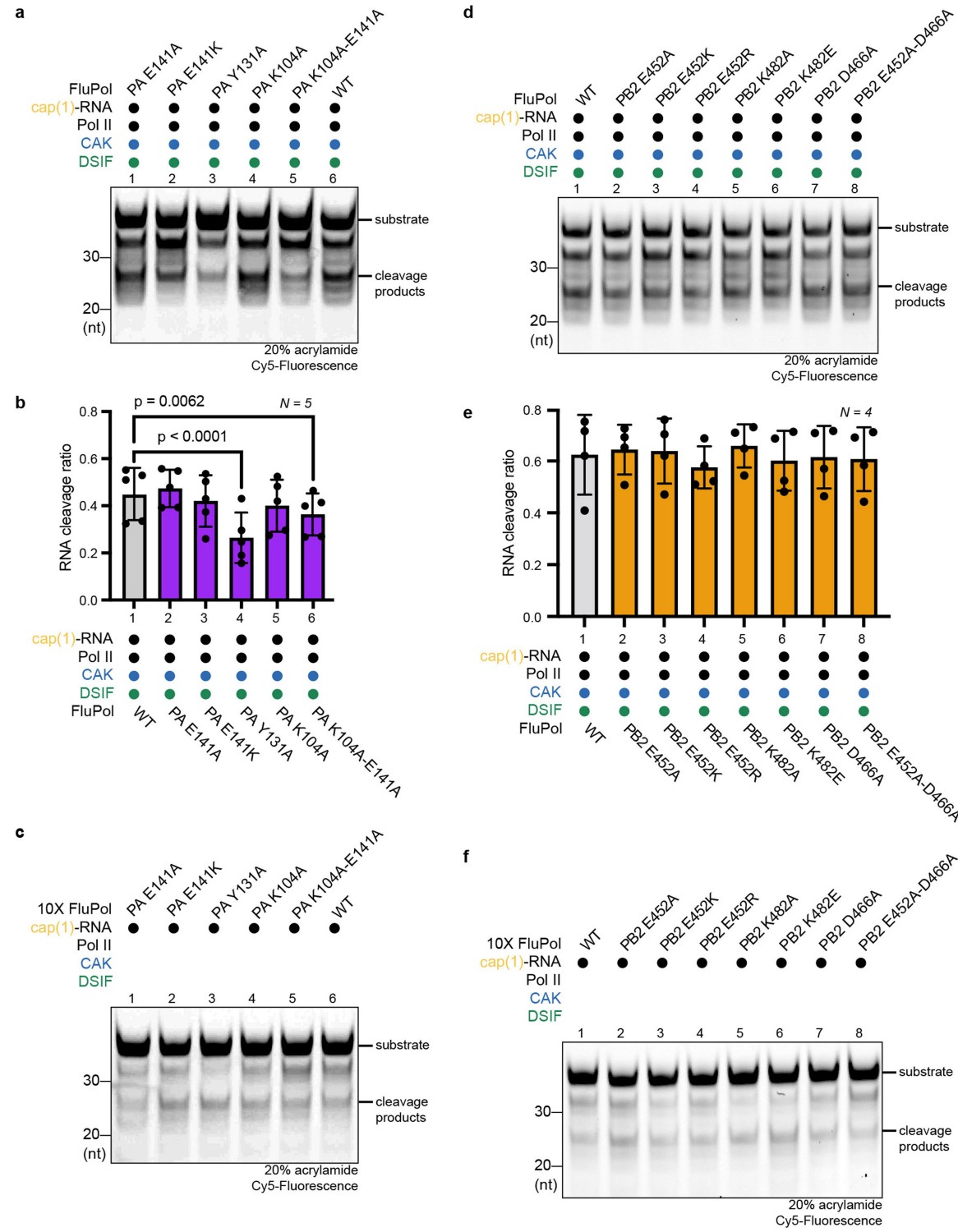

**Extended Data Fig. 6 | In vitro effects of FluPol mutations involved in the interface to the Pol II EC. a-c**, Effect of mutation of PA on the endonuclease activity of FluPol in vitro. **a**, Representative example for the denaturing urea PAGE of an endonuclease assay of FluPol interface 1 mutants using the EC as a substrate. The substrate and the product bands are labeled. **b**, The fraction of cleaved RNA (intensity of cleaved product divided by intensity of all bands, see Extended Data Fig. 1c) is shown in dependence of the mutant. Each point reflects one experimental replicate (n = 5), shown are mean ± s.d. Significance p-values were calculated using a linear mixed-effects model (substrate as a fixed effect, experimental replicate as a random effect, two-sided, no multiple testing correction). Comparison of FluPol WT and FluPol PA Y131A or FluPol PA K104A, E141A have p-values of <0.0001 or 0.0062. **c**, Representative example

for the denaturing urea PAGE of an endonuclease assay using the RNA only as a substrate shows similar activity of FluPol using ten times the amount of FluPol compared to the reactions in **a**. **d-f**, Effect of mutation of PB2 on the endonuclease activity of FluPol in vitro. **d**, Representative example for the denaturing urea PAGE of an endonuclease assay of FluPol interface 2 mutants using the EC as a substrate. The substrate and the product bands are labeled. **e**, The fraction of cleaved RNA (intensity of cleaved product divided by intensity of all bands, see Extended Data Fig. 1c) is shown in dependence of the mutant. Each point reflects one experimental replicate (n = 4), shown are mean ± s.d. **f**, Representative example for the denaturing urea PAGE of an endonuclease assay using the RNA only as a substrate shows similar activity of FluPol using ten times the amount of FluPol compared to the reactions in **d**.

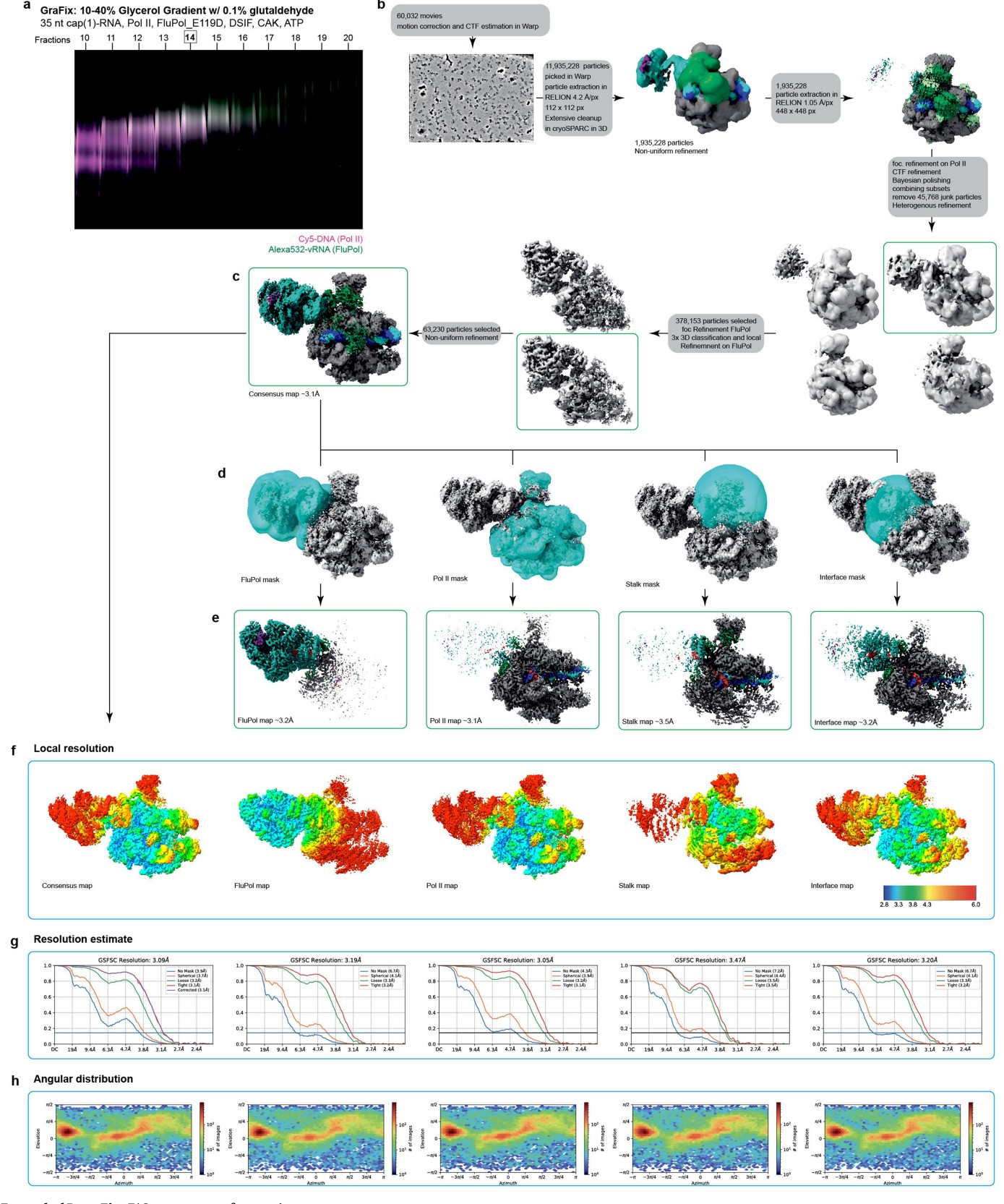

**Extended Data Fig. 7** | See next page for caption.

**Extended Data Fig. 7 | Data acquisition and processing of the post-cleavage complex. a**, Fluorescence scan of the native PAGEs analyzing GraFix gradient fractions of the post-cleavage complex preparation (n = 1). The image is an overlay of Cy5 and ATTO532 signals. In magenta is the scan for the Cy5 channel of the labeled template DNA, and in green is the scan for the ATTO532 channel of the labeled 3′ vRNA. Highlighted fractions were combined and used for cryo-EM analysis. **b**, Flowchart illustrating the key steps of the processing pipeline for obtaining the post-cleavage FluPol-Pol II-DSIF EC structure. **c**, Consensus density map of the post-cleavage FluPol-Pol II-DSIF EC colored by underlying protein components. **d**,**e**, Masks used for focused refinements and their resulting maps. **f**, Local resolution of consensus and focus refinement maps. Local resolution estimations were performed in RELION. **g**, Gold-standard Fourier shell correlation plots of the consensus and focused maps. **h**, Angular distribution of the consensus refinement and focused refinements as plotted by cryoSPARC.

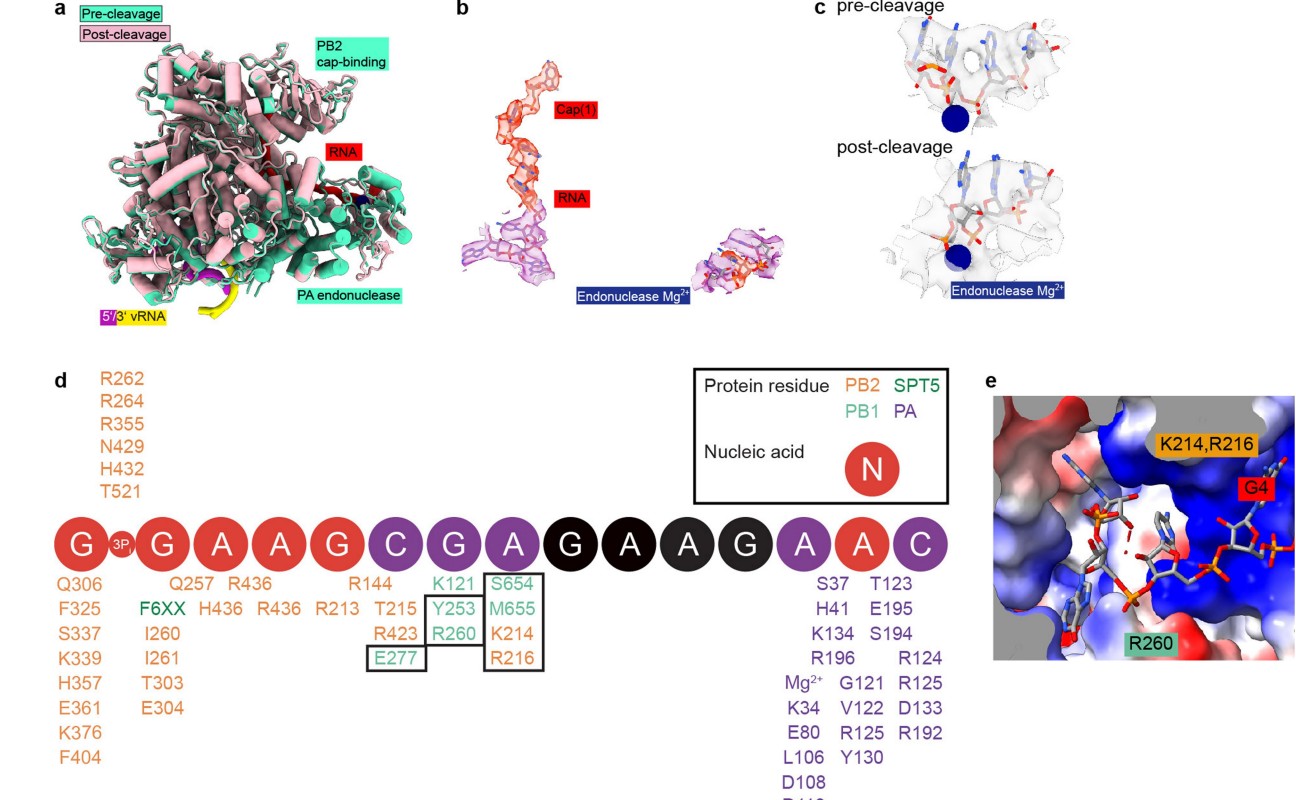

**Extended Data Fig. 8 | Features of the post-cleavage complex. a**, Comparison of pre- and post-cleavage FluPol structures. Pre-cleavage is shown in turquoise, and post-cleavage is in pink. **b**, Cryo-EM density for the RNA within FluPol between the PB2 cap-binding and PA endonuclease domains. The density is colored by thresholds. Red corresponds 0.6 and purple to 0.3. The endonuclease active site $Mg^{2+}$ is shown as a dark blue sphere. The density was restricted to areas close to the RNA in ChimeraX. The RNA is colored by heteroatom. **c**, Comparison of the Cryo-EM density of the RNA in the endonuclease for pre- and post-cleavage FluPol shows no density for the G11 nucleotide. The density was restricted to areas close to the RNA in ChimeraX using a wider cutoff. The density is shown in transparent gray and the RNA is colored by heteroatom. **d**, Schematic representation of the RNA and its interacting residues within the post-cleavage complex. The RNA is color coded by resolution using the same thresholds as in **e**, balck RNA residues are not modelled. **e**, The surface at the RNA exit of FluPol is positively charged, guiding the 3′-end of the cleaved RNA fragment into FluPol.

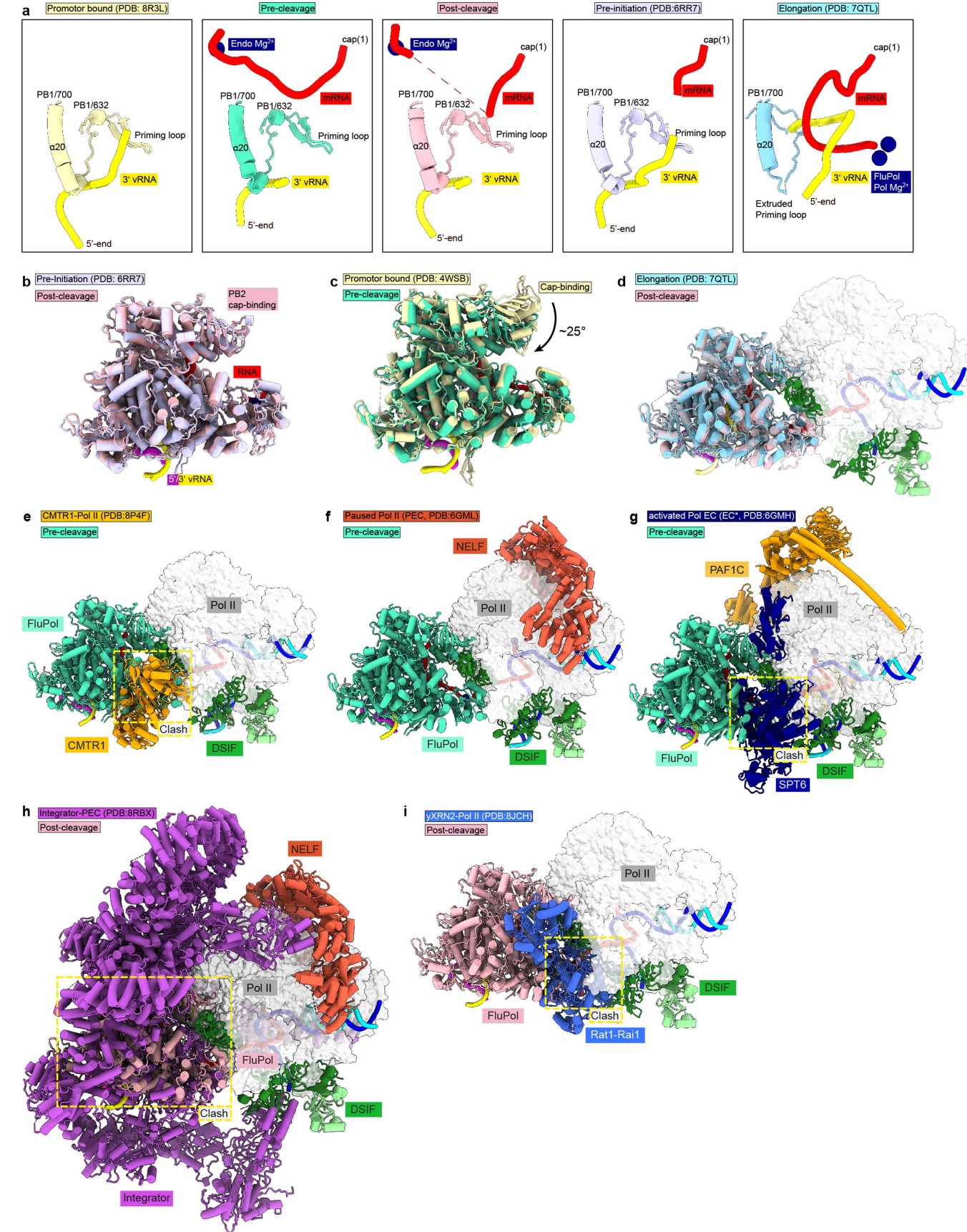

**Extended Data Fig. 9** | See next page for caption.

**Extended Data Fig. 9 | After RNA cleavage, FluPol is in a pre-initiation state and cap snatching is compatible with early Pol II elongation and pausing.** **a**, Comparison of FluPol transcription pre-initiation state (PDB: 4WSB, yellow)[48] with the pre-cleavage cap-snatching state (turquoise), post-cleavage cap-snatching state (pink), pre-initiation state (PDB: 6RR7, light purple)[44] and elongation state (PDB: 7QTL, light blue)[35]. Only the priming loop, the 3′ vRNA, and the capped RNA are depicted. **b**, Comparison of post-cleavage and pre-initiation FluPol structure shows no significant protein rearrangement. Post-cleavage is shown in pink, and pre-initiation in transparent white (PDB:6RR7)[44]. **b**, The comparison of promoter-bound FluPol with the pre-cleavage FluPol structure shows the rotation of the cap-binding domain during substrate binding. Promoter-bound FluPol is shown in yellow, and pre-cleavage in turquoise (PDB:4WSB)[44]. **d**, Superposition of an early elongating FluPol onto the post-cleavage FluPol-Pol II-DSIF EC reveals no clash between the early elongating FluPol and the Pol II EC (PDB: 7QTL, light blue)[35]. **e**, Overlay of the pre-cleavage complex (this study) and the Pol II-CMTR1 EC (PDB:8P4F) shows that FluPol clashes with CMTR1[25]. **f**, NELF binding to Pol II is compatible with the cap snatching as NELF and FluPol bind to different regions of Pol II (PDB: 6GML)[49]. **g**, FluPol clashes with SPT6 core in the activated elongation complex (PDB: 6GMH)[50]. **h**, Comparison of the post-cleavage structure with the Integrator complex bound to the paused Pol II EC (PDB: 8RBX)[51] shows a clash between FluPol and the Integrator cleavage module. **i**, Comparison of the post-cleavage structure with yeast Rat1-Rai1 (homolog of human XRN2) bound to a yeast Pol II EC (PDB: 8JCH)[75] shows a clash between FluPol and Rat1-Rai1.

**Extended Data Table 1 | Cryo-EM data acquisition, processing, and refinement statistics**

| | Pre-cleavage (EMD-50892, PDB 9FYX) | Post-cleavage (EMD-50927, PDB 9G0A) |
|---|---|---|
| **Data collection and processing** | | |
| Magnification | 81,000 | 81,000 |
| Voltage (kV) | 300 | 300 |
| Electron exposure (e–/Å²) | 39.94 | 40.0 |
| Defocus range (μm) | -0.5 to -2.0 | -0.5 to -2.0 |
| Pixel size (Å) | 1.05 | 1.05 |
| Symmetry imposed | C1 | C1 |
| Initial particle images (no.) | 6,423,874 | 11,935,228 |
| Final particle images (no.) | 369,858 | 63,230 |
| Map resolution (Å) | 3.3 | 3.1 |
| FSC threshold | 0.143 | 0.143 |
| Map resolution range (Å) | 2.74-11.4 | 2.85-11.9 |
| Map sharpening $B$ factor (Å²) | -72.3 | -60.7 |
| **Refinement** | | |
| Initial model (PDB) | 7B0Y, 5OIK,7QTL | 7B0Y, 5OIK,7QTL,7YCX |
| Model resolution (Å) | 3.3 | 3.1 |
| **Model composition** | | |
| Protein residues | 6690 | 6699 |
| Nucleic acid residues | 122 | 106 |
| Ligands | GGG:1, Zn:9, Mg:2, PO4:2 | GGG:1, Zn:8, Mg:2, PO4:3 |
| **$B$ factors (Å²)** | | |
| Protein | 98.82 | 109.27 |
| Nucleotides | 118.05 | 149.9 |
| Ligand | 60.37 | 144.58 |
| **r.m.s. deviations** | | |
| Bond lengths (Å) | 0.004 | 0.008 |
| Bond angles (°) | 0.686 | 1.168 |
| **Validation** | | |
| MolProbity score | 1.64 | 1.86 |
| Clashscore | 7.60 | 12.97 |
| Poor rotamers (%) | 0.10 | 0.08 |
| **Ramachandran plot** | | |
| Favored (%) | 96.54 | 96.39 |
| Allowed (%) | 3.46 | 3.61 |
| Disallowed (%) | 0.00 | 0.00 |

# Reporting Summary

## Statistics

For all statistical analyses, confirm that the following items are present in the figure legend, table legend, main text, or Methods section.

| n/a | Confirmed | |
|---|---|---|
| ☐ | ☒ | The exact sample size (*n*) for each experimental group/condition, given as a discrete number and unit of measurement |
| ☐ | ☒ | A statement on whether measurements were taken from distinct samples or whether the same sample was measured repeatedly |
| ☐ | ☒ | The statistical test(s) used AND whether they are one- or two-sided *Only common tests should be described solely by name; describe more complex techniques in the Methods section.* |
| ☒ | ☐ | A description of all covariates tested |
| ☐ | ☒ | A description of any assumptions or corrections, such as tests of normality and adjustment for multiple comparisons |
| ☐ | ☒ | A full description of the statistical parameters including central tendency (e.g. means) or other basic estimates (e.g. regression coefficient) AND variation (e.g. standard deviation) or associated estimates of uncertainty (e.g. confidence intervals) |
| ☐ | ☒ | For null hypothesis testing, the test statistic (e.g. *F*, *t*, *r*) with confidence intervals, effect sizes, degrees of freedom and *P* value noted *Give P values as exact values whenever suitable.* |
| ☒ | ☐ | For Bayesian analysis, information on the choice of priors and Markov chain Monte Carlo settings |
| ☒ | ☐ | For hierarchical and complex designs, identification of the appropriate level for tests and full reporting of outcomes |
| ☒ | ☐ | Estimates of effect sizes (e.g. Cohen's *d*, Pearson's *r*), indicating how they were calculated |

*Our web collection on statistics for biologists contains articles on many of the points above.*

## Software and code

Policy information about availability of computer code

| Data collection | Amersham Typhoon Scanner 3.0, Serial EM 4.0, Roche LightCycler® 480 system v1.5.0.39 |
|---|---|
| Data analysis | cryoSPARC 4.3.1, RELION 3.1, Warp 1.0.9, PHENIX 1.19.2, UCSF Chimera X-1.6, ISOLDE 1.6, Coot 0.9.6, ImageJ 2.9.0, Prism 9.4.1, Snapgene 7.0.1 |

For manuscripts utilizing custom algorithms or software that are central to the research but not yet described in published literature, software must be made available to editors and reviewers. We strongly encourage code deposition in a community repository (e.g. GitHub). See the Nature Portfolio guidelines for submitting code & software for further information.

## Data

Policy information about availability of data

All manuscripts must include a data availability statement. This statement should provide the following information, where applicable:
- Accession codes, unique identifiers, or web links for publicly available datasets
- A description of any restrictions on data availability
- For clinical datasets or third party data, please ensure that the statement adheres to our policy

The electron density reconstructions and final models were deposited with the EM Data Bank (accession codes 50892, and 50927) and the PDB (accession codes PDB 9FYX (pre-cleavage), and 9G0A (post-cleavage)).

# Research involving human participants, their data, or biological material

Policy information about studies with human participants or human data. See also policy information about sex, gender (identity/presentation), and sexual orientation and race, ethnicity and racism.

| | |
|---|---|
| Reporting on sex and gender | not applicable |
| Reporting on race, ethnicity, or other socially relevant groupings | not applicable |
| Population characteristics | not applicable |
| Recruitment | not applicable |
| Ethics oversight | not applicable |

Note that full information on the approval of the study protocol must also be provided in the manuscript.

# Field-specific reporting

Please select the one below that is the best fit for your research. If you are not sure, read the appropriate sections before making your selection.

☒ Life sciences ☐ Behavioural & social sciences ☐ Ecological, evolutionary & environmental sciences

For a reference copy of the document with all sections, see nature.com/documents/nr-reporting-summary-flat.pdf

# Life sciences study design

All studies must disclose on these points even when the disclosure is negative.

| | |
|---|---|
| Sample size | For endonuclease assays we choose the sample sizes based on the samples size of the simplified assay in the first draft. Since we observe a reproducible and statistical significant difference for the CTD phosphorylation (positive control) we assume the sample size is sufficient. For cryo-EM experiments, several million initially picked particles were collected, in line with previous published cryo-EM experiments on Pol II. Since single particle cryo-EM experiments requires averaging over thousands of particles, the sample size was sufficient. Sample sizes for cell-based assays were estimated on the basis of previous studies using similar methods and analyses that are widely published, for example see Krischuns et al., 2024. |
| Data exclusions | We excluded broken particles and poorly aligned particle classes. This is routine practice in cryo-EM to exclude such particle because they introduce noise in the reconstruction. One out of 5 qPCR experiments was taken out, because it was performed with a batch of dNTPs that was later found to be flawed. One of the endonuclease assays on FluPol mutants was removed, as an RNAse contamination was detected in one of the samples in a time point zero control sample. One of the initial 7 replicates of the endonuclease assay of DSIF dependence was ommitted because there was no activity at all, assuming a human error in the preparation of the FluPol promoter mixture. |
| Replication | Cell-based experiments using luminescence were performed with at least three independent biological replicates, with all attempts at replication being successful. Endonuclease cleavage assays were performed 5-7 times. Protein expression analyses by western-blot, analytical gel filtration, cryo-EM experiments, and elongation assays were performed once. |
| Randomization | Division of particles into random halves is automatically performed during 3D reconstruction by Relion 3.1.0. Other experiments did not involve randomization. |
| Blinding | Blinding is not applicable for this study, as group allocation is not used. |

# Reporting for specific materials, systems and methods

We require information from authors about some types of materials, experimental systems and methods used in many studies. Here, indicate whether each material, system or method listed is relevant to your study. If you are not sure if a list item applies to your research, read the appropriate section before selecting a response.

## Materials & experimental systems

| n/a | Involved in the study |
|---|---|
| ☐ | ☒ Antibodies |
| ☐ | ☒ Eukaryotic cell lines |
| ☒ | ☐ Palaeontology and archaeology |
| ☒ | ☐ Animals and other organisms |
| ☒ | ☐ Clinical data |
| ☒ | ☐ Dual use research of concern |
| ☒ | ☐ Plants |

## Methods

| n/a | Involved in the study |
|---|---|
| ☒ | ☐ ChIP-seq |
| ☒ | ☐ Flow cytometry |
| ☒ | ☐ MRI-based neuroimaging |

# Antibodies

| | |
|---|---|
| Antibodies used | Primary antibodies for analytical size exclusion:<br>rabbit anti-Strep-Tag II antibody (ab76949, Abcam)<br>rabbit anti-RPB3 polyclonal antibody (A303-771A, Bethyl)<br>Secondary for analytical size exclusion:<br>anti-rabbit antibody coupled to HRP (NA937, GE Healthcare)<br><br>Primary Ab for steady-state levels in cell-based assay:<br>Influenza A virus PA protein antibody (GTX125932, GeneTex)<br>Influenza A virus PB2 protein antibody (GTX125925, GeneTex)<br>Monoclonal Anti-α-Tubulin antibody produced in mouse (B-5-1-2, Sigma Aldrich T5168, lot number 0000089499, 1:10000)<br>Secondary Ab for steady-state levels in cell-based assay:<br>Anti-Mouse IgG (whole molecule)–Peroxidase antibody produced in rabbit (A9044, Sigma-Aldrich)<br>Anti-Rabbit IgG (whole molecule)–Peroxidase antibody produced in goat (A9169, Sigma-Aldrich) |
| Validation | The antibodies used in study has been evaluated in peer- reviewed publications. anti-RPB3 Ab (Dubbury et al., 2018), anti-PB2 Ab (Krischuns et al., 2024), anti-PA (Da Costa et al., 2015), anti-tubulin (Solinger et al., 2008)<br>anti-Tubulin<br> Validation data shown on the provider's website: upon western blot analysis, lysates from control tubulin-negative control worms do not show the expected band, while it is observed for tubulin-expressing worms (Figure 8E, Solinger et al. PLOS Genetics 2008, DOI: 10.1371/journal.pgen.1000820 )<br> anti-PA<br>A rabbit serum directed against the PA domain (residues 197 to 257) was used to reveal PA expression.<br>Validation data included in the manuscript: upon western blot analysis, lysates from control mock-transfected cells do not show the expected band migrating at 100 kDa, while it is observed for cells transfected with the PA plasmid.<br>anto-PB2<br>Validation data included in the manuscript: upon western blot analysis, lysates from control mock-transfected cells do not show the expected band migrating at 100 kDa, while it is observed for cells transfected with the PB2 plasmid.<br>anti-Strep-Tag II<br>Validation data included in the manuscript: upon western blot analysis, a band migrating at 80 kDa in FluPol (Strep-II-tagged) containing lanes. Antibody was validated by the manufacturer by Western Blot analysis used a recombinatly expressed strep-tagged protein in a lysate. A band was only observed if the strep-tagged protein was expressed.<br>anti-RPB3<br>Validation data included in the manuscript: upon western blot analysis, a band migrating at 30 kDa in Pol II containing lanes. The antibody was tested by the manufacturer using Western blot analysis and IP elutions against RPB3 using different antibodies as to pull-down RPB3. A band was observed at the expected high if antibodies against RPB3 were used during the IP, but not if a negative control antibody was used. |

# Eukaryotic cell lines

Policy information about cell lines and Sex and Gender in Research

| | |
|---|---|
| Cell line source(s) | Hi5 cells: Expression Systems, item 94-002F<br>Sf9 cells: ThermoFisher, Catalogue Number 12659017<br>Sf21 cells: Expression Systems, Item 94-003F<br>The 293T cells were purchased at ATCC (CRL-3216)<br>The MDCK cells were provided by the National Influenza Center, Paris, France. |
| Authentication | 293T cells were authentified by ATCC using STR profiling. Sex = female. Insect cell were not authenticated in-house. Sf9 cells were tested by the manufacturer by isozyme and karyotype analysis. MDCK cells were not authentified in-house. |
| Mycoplasma contamination | Not tested for insect cells. The human cell line used has been tested on a regular basis for the absence of mycoplasma, using a specific PCR detection protocol. The human cell line tested negative for mycoplasma contamination |
| Commonly misidentified lines<br>(See ICLAC register) | No commonly misidentified cell lines were used in the study. |

# Plants

**Seed stocks**

*Report on the source of all seed stocks or other plant material used. If applicable, state the seed stock centre and catalogue number. If plant specimens were collected from the field, describe the collection location, date and sampling procedures.*

**Novel plant genotypes**

*Describe the methods by which all novel plant genotypes were produced. This includes those generated by transgenic approaches, gene editing, chemical/radiation-based mutagenesis and hybridization. For transgenic lines, describe the transformation method, the number of independent lines analyzed and the generation upon which experiments were performed. For gene-edited lines, describe the editor used, the endogenous sequence targeted for editing, the targeting guide RNA sequence (if applicable) and how the editor was applied.*

**Authentication**

*Describe any authentication procedures for each seed stock used or novel genotype generated. Describe any experiments used to assess the effect of a mutation and, where applicable, how potential secondary effects (e.g. second site T-DNA insertions, mosiacism, off-target gene editing) were examined.*

