## [Peer Review File · Nature]

Mechanism of Co-Transcriptional Cap-Snatching by Influenza Polymerase

Corresponding Author: Dr Christian Dienemann

Version 0:

Reviewer comments:

Referee #1

(Remarks to the Author)

Influenza A virus cap-snatching and transcription initiation are the first activities that the viral RNA polymerase performs upon entering the nucleus of the host cells. The cap-snatching was first described in 1979-1980 and the association of the viral RNA polymerase with active chromatin in 1982. Since then, numerous labs have studied the biochemical and structural steps of cap-cleavage and transcription initiation process, and the binding of the viral RNA polymerase to host cell RNA polymerase II (Pol II) and its co-factors. However, a direct image of the viral enzyme preying on the host transcription machinery has so far remained elusive.

In this fantastic manuscript, virology and structural biology labs use cryoEM and biochemistry to reveal two steps of influenza A virus cap-snatching. The structures capture beautifully how the PB2 cap-binding domain of the influenza A virus RNA polymerase inserts itself between subunits of Pol II to get access to the emerging capped RNA. The study also reveals key residue interactions and the movements that the host and viral RNA proteins undergo during the cap snatching process. Finally, the study provides insight into when in the carefully regulated host transcription process the influenza A virus RNA polymerase has an opportunity to prey on emerging capped RNAs.

The authors use Pol II purified from pig cells and a H7N9 influenza virus RNA polymerase purified from insect cells. A mutation was introduced into the endonuclease domain of the RNA polymerase to slow down its cap cleavage activity. In addition, the authors cleverly played with the magnesium ion concentration to capture two stages of the cap-snatching process. The authors also use biochemical experiments to largely confirm that interactions between the viral RNA polymerase and a phosphorylated Pol II CTD and a cap-1 structure are critical for influenza virus RNA polymerase cap-snatching. The authors also show that addition of elongation factor DSIF increases cap-snatching.

Overall, the new structures are of great importance for our understanding of influenza A virus cap snatching. However, there are quite a few points in the manuscript where the authors seem to be going through some corners a little fast. These points should be addressed.

Specific comments:

Line 122. Clarify in the text and legends when the wildtype enzyme or the E119D mutant was used in the in vitro assays. At the moment, one needs to puzzle it together from the methods.

Line 119-139. Were the described enzymatic experiments performed using purified complexes or by mixing the components at stoichiometric ratios or something else? Please clarify.

Fig. 1d, EDF 1 panel b. The reaction describes the cleavage of a 35-nt long cy5-labeled capped RNA that is hybridized to a DNA oligo. After cleavage, the fluorescent signal runs at ~30 or ~24 nt. The cy5 label is attached to the 3' end (please add this information to the methods; the sequence is written in IDT format and stating the label near the 5' end is confusing), so

the authors are not visualizing the primer, but what is left over of the nascent RNA. Subtracting the product size from the substrate length suggests that the latter capped primer is likely 11 nt long, which matches the location of the AG cleavage site in the provided RNA sequence. However, the other capped primer is ~5 nt long. In turn, this suggests that cleavage on this hybrid substrate is different from what is typically observed, which ought to be mentioned (e.g. at line 271 where this is ignored). Moreover, which band was quantified for Fig. 1d?

EDF 1 panel c. This appears to be autoradiography/phosphorimaging data, not scintillation. The latter relies on a solvent/scintillation fluid and light.

Lines 124-126, 135-136, 269. The finding that CTD binding does not improve endonuclease activity is presented as fact or a key finding, but it contradicts the findings by others. For instance, Martinez-Alonso et al (JVI) and Serna Martin et al (Mol Cell) reported an increase in viral RNA endonuclease and transcription activity upon Ser5-CTD addition. There can be many reasons for differences, but at least it should be mentioned.

Line 132-133, Fig. 1d, EDF 1b and c. One of the assays measures endonuclease activity (fluorescent data presented in 1d and EFT 1b) and the other endonuclease activity + elongation (EDF 1c; capped RNA of 35 nt that is converted into a radiolabeled product of ~32 nt). None of the assays actually measures extension alone. I appreciate that such an assay may not be a fair here as there could be signal without complex formation. However, the authors could calculate the effect of DSIF on elongation if they quantify the data presented in EDF 1d and ideally perform the assay with the fluorescent primer (though they may need to change the template length since the input and product signals will be running close together).

Fig. 3 and line 218. The authors used a minigenome reporter assay to confirm that the observed interactions are important for viral transcription. The assay used relies on both transcription and replication to produce a luciferase signal. A similar reduction in signal can be seen if a replication mutant is present. Hence, the assay is far from ideal to support the author's conclusions that these residues are important for 1) transcription (alone) and 2) for establishing the interaction observed in the cryo-EM structures. The authors should use a more specific readout to measure the viral mRNA levels in cells (e.g. using primer extension or qPCR to plot m/v ratios as in Lukarska et al Nature) and they should confirm that the PA or PB2 domains interact less with Pol II using immunoprecipitation or similar.

Line 259-261, 271-273. While no rotation may be observed in ~5% of the data and rotation not required to get the 3' end of the capped primer into the nascent strand exit channel, rotation may be required to establish stable base pairing between the primer and the template. The latter is not seen in the data, so the authors are over-interpreting what they see. Moreover, if the cleaved primer is relatively short (e.g. 8-9 nt as observed in both gel and sequencing data; and some strains show a bias towards short primers), would there be enough ssRNA to get the 3' end into the active site to stably base pair with the 3' end of the template without rotation. The authors only resolve 7 nt in this conformation, so probably transcription cannot start without rotation? It may be useful to discuss the above and also be thoughtful of any interactions between the influenza RNA polymerase and Pol II that need to be broken before between the cap binding domain can rotate and viral transcription can start.

Line 272. The authors propose that the 3' end of the capped primer 'swings' towards the RNA polymerase active site. Such a pendulum-like behavior implies that the 3' end may also be swinging the wrong way over half of the time and that the influenza A virus RNA polymerase is fumbling with the 3' end like comedian with a wet bar of soap. I assume the authors just phrased this casually. Perhaps they can take into account the positive charge near the exit channel and that this may help guide the 3' end in, swinging or not?

Supplemental movie: at around 1 min 40, the 3' end of the vRNA template (yellow) appears to fold on itself in the entry channel instead of base-pair with the red primer. At the same time, the 3' end of the red primer appears to be shying away from the template. Tracking the 5' and 3' ends of the template, it appears that the 5' end ends up forming base pairs with the primer. I assume this is an animation/morphing artefact? In any case, this sequence of depicted events appears to contradict previous structures from the Cusack lab as well as biochemical sense (e.g. Kouba et al, and Wandzik et al).

Referee #2

(Remarks to the Author)

This manuscript reported two critical structures of FluPol for revealing the mechanism of its cap-snatching mechanism for viral transcription.

The first important structure is that of FluPol in complex with capped-RNA, Pol II and DSIF. The complex of capped RNA, Pol II and DSIF represents the early elongation complex of host transcription. There are intimate interactions between FluPol and Pol II-DSIF. The PA endonuclease directly interacts with DSIF, while the PB2 cap binding domain directly interacts with Pol II, the latter is dependent on phosphorylation of Pol II CTD. These interactions are required for full endonuclease activity, supported by the biochemical and mutagenesis data presented here. This complex new insights on how the transcription of FluPol by cap-snatching is tightly coordinated with the host transcription machinery. Once again, it is demonstrated that the virus evolves to suite the host functions for its own efficient replication.

The second structure showed that the cleaved capped RNA remains bound by the PB2 cap binding domain and the RNA 3' end can direct itself toward the polymerase active site of FluPol. The authors define this structure as the pre-initiation complex of viral transcription, while FluPol remains associated with Pol II-DSIF.

It is proposed that the initiation of viral transcription could occur without dissociation of FluPol from Pol II. However, the association is weakened after the capped RNA is cleaved, compared to the uncleaved RNA that has its association with

both Pol II and the PB2 cap binding domain. In addition, other host factors also compete with binding to Pol II. It would be advantageous for viral transcription if FluPol releases Pol II for another round of transcription.

A minor suggestion as the following:
Line 77, "genes"->"transcripts"

Referee #3

(Remarks to the Author)

This ms reports two cryo-EM structures of FluPol bound to PolIII and the DSIF cofactor.

Both structures were obtained with an endonuclease-impaired FluPol mutant E119D presumably to prevent the 35 nt capped RNA emerging from PolIII to be completely hydrolyzed and to allow the trapping of reaction intermediates using carefully crafted experimental procedures. One structure, obtained at low Mg concentration (0.1 mM?) captures the "pre-cleavage state" depicting the path of the capped RNA at the interface between the two polymerases (Figs 1c for a schematic view and Fig. 2 with 2a highlighting two interface regions labelled 1 and 2 and 2b the path of capped RNA). In the second structure that was obtained from averaging a very minor population of particles (line 735: 63,230 particles from 11,935,228 total particles! That is 0.5%), obtained at higher 3 mM Mg concentration, even though the overall interface is conserved compared to the precleavage state, the capped RNA has been cleaved and the 3' end is redirected towards the Flupol active site (Fig. 4).

Together these structures give mechanistic insight into a key step of influenza replication, where a 5' cap structure needed to stabilize viral RNA against hydrolysis, is acquired via "cap snatching" that is by stealing a capped host transcript synthesized by PolIII and to use it for transcription of viral mRNA that can subsequently be translated into viral proteins. The structural work is complemented by minigenome studies where residues at the interface region 1 and 2 have been targeted for mutations and the impact on FluPol activity was measured (Fig. 3).

Overall, the work is potentially of significance as cap snatching was discovered 43 years ago (ref 1) and the precise mechanism for it has remained largely elusive (although groundbreaking related work by the Cusack group has already been reported eg reviewed in ref 15). Influenza is still a major potential threat of pandemics, and this work could indeed help the design of antiviral inhibitors targeting Flupol-PolIII interfaces.

Major issues.

1-As mentioned above, one main issue is the reliability/validity of the so called "post-cleavage" state given that it is derived from a very minor subpopulation and from only one fraction 14 in ED Fig. 5a. Likewise, ED Fig. 1d shows that with 3.0 mM Mg, the major band remains the 35 nt capped RNA (as in the 0.1 mM fraction) and only a very minor amount of RNA is cleaved into two fragments (left lane). It would have been useful to try to analyze the identity of these fragments and correlate with what is observed in the postcleavage structure. Hence there is the possibility that what is called here the postcleavage state is only some intermediate that follows immediately the precleavage state. These should be at least discussed as possible limitations of the study.

2- The quality of the Figures should be markedly improved (eg Fig 2 and 4). Magnified views of capped RNA should be given and in connection with the possible criticism raised in 1-, the orientation of capped RNA in both the precleavage and even more in the postcleavage should be documented and supported by clear electron density maps documenting the proper assignment of bases. In this respect having a 1D sequence of capped RNA clearly highlighting (with color code) what is built, what is disordered and the interactions with the proteins would help. This could be a modification of ED Fig 3.

3-ED Fig. 3. Poor figure quality. Color choice? Comparison between the present structure of flupol-polIII and prior CTD-polIII complex (ref 2, 42) needs to be shown better. Give r.m.s deviations.

4-In addition, the sentence line 246 "density discontinues abruptly" should be accompanied by electron density to document whether this break matches with the endonuclease active site and in addition temperature factors of Mg ions should be given (they should be lower for the postcleavage compared to precleavage?).

5- Some of the writing could be improved/be more specific
eg line 179, "phosphorylation of serine residues" which one?

Line 176: "an interaction as proposed before" not very clear what was proposed.

6-For the mutation experiments in Fig. 3, given the uncertainty, can the authors envision validating their hypothesis (lines 207,208 key salt bridge between E141 and K627 from DSIF) which has a 8-10 fold impact on FluPol activity by swapping the charges to restore activity eg by a E141K, K627E double mutant and check on the minigenome assay?

Other issues:

1-Flupol mutant E119D is used throughout but its precise location is never shown in the structures including the Endonuclease domain... (unless I missed that).

2-Line 97: 96% identity: Vague formulation! Presenting 96% sequence identity? Pol II is a protein complex...

3-Line 115: nearly no effect

4-Line 129-130: The authors should consider to include a negative control (e.g. RNA + Pol II + CAK + DSIF) to confirm that the increased activity is dependent on FluPol and exclude the possibility that the 2-fold increase of RNA cleavage observed in the presence of the elongation factor DSIF is not caused by any carry-over of RNase contamination from the DSIF sample

itself.

5-Line 132-133: The authors should provide a quantitative analysis of the signal intensity from ED Figure 1c for a better interpretation of the data.

Actually from the data in ED Figure 1b and 1c, it is not clear whether the RNA extension activity can be correlated with the activity of the endonuclease activity.

6-Fig 1: Swap data in a. and b. panels. A possibility is to present first in a. the SEC data and secondly confirm the presence of the protein subunits by Western-blot in b.

7- Line 364-365: A paired parametric t-test is incorrect. The authors should perform an UNPAIRED parametric t-test when they want to compare the statistical significance between two conditions (unpaired conditions).

A paired parametric t-test is used when two measurements are conducted onto the same object before and after an experimental treatment. E.g. measure the body temperature before and after paracetamol intake.

However, in the present case the authors apply two different treatments (complex 3 vs complex 4) on their labelled RNA. So there is no connections, the t-test should be unpaired.

8-ED Table 1: precleavage B factor nucleotides 1117.71 A2, I hope it is a typo...

Version 1:

Reviewer comments:

Referee #1

(Remarks to the Author)

During influenza A virus infection, RNA polymerase II binding and cap-snatching are the first activities of viral RNA polymerase. The cap-snatching activity by the RNA polymerase was first described in 1979-1980 and the association of the viral RNA polymerase with active euchromatin in 1982. Since then, numerous labs have studied the biochemical and structural steps of cap-cleavage and transcription initiation process, as well as the binding of the viral RNA polymerase to host cell RNA polymerase II (Pol II) and its co-factors. However, a direct image of the viral enzyme shouldering its way into the host transcription machinery to steal the ends of nascent mRNAs has so far remained elusive.

In this revised manuscript, the authors describe two states of the cap-snatching process: a pre-cleavage state, in which the influenza virus RNA polymerase has bound RNA polymerase 2 and triggered rearrangements in the host complex to gain access to the emerging nascent capped RNA, and a post-cleavage state. The authors confirm the validity of the structures, interactions, and the critical components of the cap-snatching process using stepwise biochemical assays, clever use of magnesium ion concentrations, and an impressive array of point and combinatory mutations. The data fully support the interpretation of the structures and the presented conclusions are clear and nuanced, in line with the power of the experiments and controls.

Overall, the study confirms, extends and in some places corrects previously observed models or interactions and provides a near complete picture of one of the first, and arguably most important, enzymatic steps in an influenza virus infection. The authors discuss how the study reveals that the influenza A virus RNA polymerase likely interacts with Pol II after methylation of the cap of the nascent mRNA. The authors also touch upon the role of NELF, which was recently described to be important for influenza virus transcription (<https://pubmed.ncbi.nlm.nih.gov/39605461/>) and downstream immune signaling in a preprint, and when or if the viral enzyme may dissociate from the Pol II complex. This discussion is clearly written and it nicely places the structures in a broader context, paving the way for future studies.

This manuscript presents not only a thorough study but a major advance in our understanding of influenza virus infections, and likely virology in general. I fully endorse the manuscript.

Minor point:

Line 322: "flexibly cap-binding domain" is likely "flexible cap-binding domain"

Referee #3

(Remarks to the Author)

The authors have addressed reviewer's concerns well.

Reviewer 1:

Influenza A virus cap-snatching and transcription initiation are the first activities that the viral RNA polymerase performs upon entering the nucleus of the host cells. The cap-snatching was first described in 1979-1980 and the association of the viral RNA polymerase with active chromatin in 1982. Since then, numerous labs have studied the biochemical and structural steps of cap-cleavage and transcription initiation process, and the binding of the viral RNA polymerase to host cell RNA polymerase II (Pol II) and its co-factors. However, a direct image of the viral enzyme preying on the host transcription machinery has so far remained elusive.

In this fantastic manuscript, virology and structural biology labs use cryoEM and biochemistry to reveal two steps of influenza A virus cap-snatching. The structures capture beautifully how the PB2 cap-binding domain of the influenza A virus RNA polymerase inserts itself between subunits of Pol II to get access to the emerging capped RNA. The study also reveals key residue interactions and the movements that the host and viral RNA proteins undergo during the cap snatching process. Finally, the study provides insight into when in the carefully regulated host transcription process the influenza A virus RNA polymerase has an opportunity to prey on emerging capped RNAs.

The authors use Pol II purified from pig cells and a H7N9 influenza virus RNA polymerase purified from insect cells. A mutation was introduced into the endonuclease domain of the RNA polymerase to slow down its cap cleavage activity. In addition, the authors cleverly played with the magnesium ion concentration to capture two stages of the cap-snatching process. The authors also use biochemical experiments to largely confirm that interactions between the viral RNA polymerase and a phosphorylated Pol II CTD and a cap-1 structure are critical for influenza virus RNA polymerase cap-snatching. The authors also show that addition of elongation factor DSIF increases cap-snatching.

Overall, the new structures are of great importance for our understanding of influenza A virus cap snatching. However, there are quite a few points in the manuscript where the authors seem to be going through some corners a little fast. These points should be addressed.

Specific comments:

1. Line 122. Clarify in the text and legends when the wildtype enzyme or the E119D mutant was used in the in vitro assays. At the moment, one needs to puzzle it together from the methods.
2. Line 119-139. Were the described enzymatic experiments performed using purified complexes or by mixing the components at stoichiometric ratios or something else? Please clarify.

We thank the reviewer for helping clarify the biochemistry section. In the Results section text and legends, we now included the information on whether we used WT protein or FluPol PA E119D (Point 1). We also expanded the explanation of the endonuclease assay to help the reader understand the corresponding results (Point 2).

3. Fig. 1d, EDF 1 panel b. The reaction describes the cleavage of a 35-nt long cy5-labeled capped RNA that is hybridized to a DNA oligo. After cleavage, the fluorescent signal runs at ~30 or ~24 nt. The cy5 label is attached to the 3' end (please add this information to the methods; the sequence is written in IDT format and stating the label near the 5' end is confusing), so the authors are not visualizing the primer, but what is left over of the nascent RNA.

We thank the reviewer for pointing this out. We included the information about the cap structure and the Cy5 label in the RNA sequence in the Methods section. We also included in the method section that we visualise the RNA that remains with Pol II after cleavage by FluPol (see point 2).

4. Subtracting the product size from the substrate length suggests that the latter capped primer is likely 11 nt long, which matches the location of the AG cleavage site in the provided RNA sequence. However, the other capped primer is ~5 nt long. In turn, this suggests that cleavage on this hybrid substrate is different from what is typically observed, which ought to be mentioned (e.g. at line 271 where this is ignored).

We included the information about the additional cleavage product in the description of the results. However, we do not think that this additional cleavage product is an artefact specific to the RNA used in the assay. In fact, we performed the same set of assays using the same RNA:DNA substrate but with FluPol from B/Memphis (see Reviewer Figure 1). In this case, we did not observe the additional cleavage product. Therefore, we concluded that the additional product can be attributed to the H7N9 strain. We think this band represents a cleavage event without cap binding by PB2.

Reviewer Figure 1: **FluPol of Influenza B/Memphis does not produce the unspecific cleavage product observed with H7N9.** Representative example for the denaturing urea PAGE of an endonuclease assay. The substrate and the product bands are labelled. Lane 1 contains some spillover from the marker lane.

5. Moreover, which band was quantified for Fig. 1d?

We quantified all product bands and divided them by the sum of the product and substrate bands. To help visualise this, we included an extended data figure panel (ED Figure 1c) illustrating the quantification.

6. EDF 1 panel c. This appears to be autoradiography/phosphorimaging data, not scintillation. The latter relies on a solvent/scintillation fluid and light.

We thank the reviewer for pointing out this mistake; we have made the necessary changes to the description.

7. Lines 124-126, 135-136, 269. The finding that CTD binding does not improve endonuclease activity is presented as fact or a key finding, but it contradicts the findings by others. For instance, Martinez-Alonso et al (JVI) and Serna Martin et al (Mol Cell) reported an increase in viral RNA endonuclease and transcription activity upon Ser5-CTD addition. There can be many reasons for differences, but at least it should be mentioned.

We agree with the reviewer and have redone the endonuclease assay to accommodate an important control requested by reviewer #3. In this process, we optimised the assay to accommodate for batch variation of FluPol activity. We now observe the expected increase in cleavage rate upon the addition of the CAK CTD kinase. We changed the description and cited the corresponding literature.

8. Line 132-133, Fig. 1d, EDF 1b and c. One of the assays measures endonuclease activity (fluorescent data presented in 1d and EFT 1b) and the other endonuclease activity + elongation (EDF 1c; capped RNA of 35 nt that is converted into a radiolabeled product of ~32 nt). None of the assays actually measures extension alone. I appreciate that such an assay may not be a fair here as there could be signal without complex formation. However, the authors could calculate the effect of DSIF on elongation if they quantify the data presented in EDF 1d and ideally perform the assay with the fluorescent primer (though they may need to change the template length since the input and product signals will be running close together).

We agree that it is very unfortunate that there is currently no fluorescence-based system to measure FluPol transcription elongation rate alone. This is because the 5'-end, which is conventionally used to fluorescently label the RNA primer, is required to have a 5'cap. However, we have quantified the two replicates of the radio-labelled elongation experiment that we already had. We have normalised the data onto the sample containing both phosphorylation by CAK and DSIF. The results are shown in Reviewer Figure 2. Comparing the sample without CAK and DSIF with the sample with CAK and DSIF revealed a ~2-fold difference, similar to what has been observed in the endonuclease assay. This suggests that the difference in the radioactive elongated primer is mainly caused by the difference in cleavage activity. However, we would like to exclude the quantification of the bands from the radioactive assays for the following reasons:

1. The main point of this experiment is to show that the increased endonuclease activity will produce usable RNA primers for elongation and not to compare the effect of DSIF/CAK on cleavage and elongation rate. We have changed the wording in the Results section to clarify this.
2. Our radioactivity assays have only a modest resolution, so quantification of band intensities will result in poorer data quality compared to the endonuclease assay.
3. By including the quantification, readers will compare the effect size in the radioactivity- and fluorescence-based assays. However, these can only be partially correlated, as protein concentrations and buffer composition slightly differ between experimental setups due to technical requirements.

Reviewer Figure 2: **Quantification of elongation product by FluPol in phosphoimaging data.**

9. Fig. 3 and line 218. The authors used a minigenome reporter assay to confirm that the observed interactions are important for viral transcription. The assay used relies on both transcription and replication to produce a luciferase signal. A similar reduction in signal can be seen if a replication mutant is present. Hence, the assay is far from ideal to support the author's conclusions that these residues are important for 1) transcription (alone) and 2) for establishing the interaction observed in the cryo-EM structures. The authors should use a more specific readout to measure the viral mRNA levels in cells (e.g. using primer extension or qPCR to plot m/v ratios as in Lukarska et al., Nature) and they should confirm that the PA or PB2 domains interact less with Pol II using immunoprecipitation or similar.

We thank the reviewer for the comment and agree that the luciferase assay lacks specificity to determine whether the observed effects are transcription-dependent. Consequentially, we provide another two lines of evidence that these interfaces are important for viral transcription.

We performed strand-specific RT-qPCR to quantify the different RNA species as suggested by the reviewer. The results are attached in **ED 5c,d**. Charge reversal mutants in PB2 E452 and D466 have a reduced m/v ratio in the RT-qPCR. Consistently, recombinant viruses harbouring these mutations showed an attenuated phenotype and/or genetic instability.

We have further cloned, expressed and purified FluPol protein harbouring some of the mutations used for the luciferase assay. Additionally, we have mutated the interacting residues/regions in DSIF. We have performed endonuclease assays with these mutated proteins as described for wild-type FluPol (**ED 5c-h**). Mutations in the interface between FluPol and DSIF, which stimulates the endonuclease activity (Fig. 1d), reduce the endonuclease activity on a Pol II + CAK + DSIF substrate. However, mutations in the interface to the Pol II core did not lead to a decreased endonuclease activity. This is in accordance with the lack of stimulation by the unphosphorylated core of Pol II alone (Fig. 1d). Overall, we conclude that the interfaces between PA and DSIF, and between PB2 and the Pol II surface, are crucial for efficient cap-snatching and viral transcription. We decided to not follow the reviewer's suggestion to investigate complex formation of mutant FluPol with the Pol II elongation complex by IP or similar experiments. Because the nature of cap snatching complex formation is highly multivalent (CTD, RNA, DSIF, Pol II), we think that we would observe a high number of indirect interactions.

10. Line 259-261, 271-273. While no rotation may be observed in ~5% of the data and rotation not required to get the 3' end of the capped primer into the nascent strand exit channel, rotation may be required to establish stable base pairing between the primer and the template. The latter is not seen in the data, so the authors are over-interpreting what they see. Moreover, if the cleaved primer is relatively short (e.g. 8-9 nt as observed in both gel and sequencing data; and some strains show a bias towards short primers), would there be enough ssRNA to get the 3' end into the active site to stably base pair with the 3' end of the template without rotation. The authors only resolve 7 nt in this conformation, so probably transcription cannot start without rotation? It may be useful to

discuss the above and also be thoughtful of any interactions between the influenza RNA polymerase and Pol II that need to be broken before between the cap binding domain can rotate and viral transcription can start.

We thank the reviewer for highlighting this statement. Both the pre-cleavage and post-cleavage structures already have the cap-binding domain configuration of a pre-initiating FluPol (compare our structures with PDB:6RR7, ED. **Fig 9b**). We conclude that the rotation of the cap-binding domain either occurs during the binding of the cap(1) structure or not at all. It may just be a stabilisation of the otherwise flexibly orientated cap-binding domain in the pre-initiation conformation upon binding the Pol II-DSIF substrate. This order of events also aligns with the X-ray and cryo-EM structures from Kouba *et al.* 2019 and Fan *et al.* 2019, showing the cap-binding domain rotated inwards at the pre-initiation and initiation states. We have modified the text to clarify this point and included a comparison of the promoter-bound FluPol (in the absence of capped RNA, PDB:4WSB) and the pre-cleavage FluPol in the cap-snatching complex, highlighting the cap-binding domain position before and after binding of the 5'-cap (**ED Figure 9c**). Note, however that the outward position of the cap-binding domain has only ever been observed in one crystal form and could be a crystal packing artefact.

We also thank the reviewer for highlighting the importance of RNA length in proper initiation. Kouba *et al.* 2019 used a 15 nt long capped RNA and, using a template with an extra 3 nucleotides at the 3' end, observed a 5 nt long RNA: RNA hybrid. We have estimated that in our work, the major cleavage product has a length of 11 nt. We, therefore, conclude that an 11 nt primer is surely long enough to reach the 3'vRNA and that at least the last nucleotide base pairs. In line with this, the cleavage products can be used for primer extension by FluPol (**ED Figure 1d**). Despite our clear observations here, we agree that additional intermediate steps may exist that remain to be studied beyond the scope of this study.

The second question is whether shorter primers can reach the polymerase active site. In the initiation model of Kouba *et al.* 2019, mRNA nucleotide 11 is located at the RNA exit, but the RNA exhibits one turn. Assuming this turn does not occur on shorter RNA primers, this might allow for enough space that this base pairing occurs, although the RNA would have to be stretched, particularly for an 8-9mer. This initially weak base pairing could then be supported by a prime and reanneal mechanism as described by Velthuis and Oymans 2018.

11. Line 272. The authors propose that the 3' end of the capped primer 'swings' towards the RNA polymerase active site. Such a pendulum-like behavior implies that the 3' end may also be swinging the wrong way over half of the time and that the influenza A virus RNA polymerase is fumbling with the 3' end like comedian with a wet bar of soap. I assume the authors just phrased this casually. Perhaps they can take into account the positive charge near the exit channel and that this may help guide the 3' end in, swinging or not?

We agree with the reviewer that the driving force of RNA reorganisation is an important point and that 'directed' is more appropriate than 'swings'. We addressed this point by including an additional figure panel (**ED Figure 7h**) highlighting the positive surface charge at the end of the RNA exit channel, where the RNA primer is directed. We identified the positively charged residues and named them in the main text.

12. Supplemental movie: at around 1 min 40, the 3' end of the vRNA template (yellow) appears to fold on itself in the entry channel instead of base-pair with the red primer. At the same time, the 3' end of the red primer appears to be shying away from the template. Tracking the 5' and 3' ends of the template, it appears that the 5' end ends up forming base pairs with the primer. I assume this is an animation/morphing artefact? In any case, this sequence of depicted events appears to contradict previous structures from the Cusack lab as well as biochemical sense (e.g. Kouba et al, and Wandzik et al).

We thank the reviewer for noticing this. The 3' vRNA does not fold on itself during initiation, although the cartoon's representation and the viewing angle might let it appear like it is folding on itself. The observed bending relates to the formation of the double-stranded RNA helix in the annealing step. We modified the movie such that more of the 3' fragment is visible

Ref 2:

This manuscript reported two critical structures of FluPol for revealing the mechanism of its cap-snatching mechanism for viral transcription.

The first important structure is that of FluPol in complex with capped-RNA, Pol II and DSIF. The complex of capped RNA, Pol II and DSIF represents the early elongation complex of host transcription. There are intimate interactions between FluPol and Pol II-DSIF. The PA endonuclease directly interacts with DSIF, while the PB2 cap

binding domain directly interacts with Pol II, the latter is dependent on phosphorylation of Pol II CTD. These interactions are required for full endonuclease activity, supported by the biochemical and mutagenesis data presented here. This complex new insights on how the transcription of FluPol by cap-snatching is tightly coordinated with the host transcription machinery. Once again, it is demonstrated that the virus evolves to suite the host functions for its own efficient replication.

The second structure showed that the cleaved capped RNA remains bound by the PB2 cap binding domain and the RNA 3' end can direct itself toward the polymerase active site of FluPol. The authors define this structure as the pre-initiation complex of viral transcription, while FluPol remains associated with Pol II-DSIF.

It is proposed that the initiation of viral transcription could occur without dissociation of FluPol from Pol II. However, the association is weakened after the capped RNA is cleaved, compared to the uncleaved RNA that has its association with both Pol II and the PB2 cap binding domain. In addition, other host factors also compete with binding to Pol II. It would be advantageous for viral transcription if FluPol releases Pol II for another round of transcription.

A minor suggestion as the following:

Line 77, "genes"->"transcripts"

We thank the reviewer for the kind words and helpful comments. We have decided for the word genes in this context as the cited source used ChIP-qPCR as a method to study FluPol cap-snatching relative to the position in the gene.

We also highlighted the importance of the dissociation of FluPol from Pol II in the discussion, as this might not be obvious to the reader.

Referee #3 (Remarks to the Author):

This ms reports two cryo-EM structures of FluPol bound to PolIII and the DSIF cofactor. Both structures were obtained with an endonuclease-impaired FluPol mutant E119D presumably to prevent the 35 nt capped RNA emerging from PolIII to be completely hydrolyzed and to allow the trapping of reaction intermediates using carefully crafted experimental procedures. One structure, obtained at low Mg concentration (0.1 mM?) captures the "pre-cleavage state" depicting the path of the capped RNA at the interface between the two polymerases (Figs 1c for a schematic view and Fig. 2 with 2a highlighting two interface regions labelled 1 and 2 and 2b the path of capped RNA). In the second structure that was obtained from averaging a very minor population of particles (line 735: 63,230 particles from 11,935,228 total particles! That is 0.5%), obtained at higher 3 mM Mg concentration, even though the overall interface is conserved compared to the precleavage state, the capped RNA has been cleaved and the 3' end is redirected towards the Flupol active site (Fig. 4).

Together these structures give mechanistic insight into a key step of influenza replication, where a 5' cap structure needed to stabilize viral RNA against hydrolysis, is acquired via "cap snatching" that is by stealing a capped host transcript synthesized by PolIII and to use it for transcription of viral mRNA that can subsequently be translated into viral proteins. The structural work is complemented by minigenome studies where residues at the interface region 1 and 2 have been targeted for mutations and the impact on FluPol activity was measured (Fig. 3).

Overall, the work is potentially of significance as cap snatching was discovered 43 years ago (ref 1) and the precise mechanism for it has remained largely elusive (although groundbreaking related work by the Cusack group has already been reported eg reviewed in ref 15). Influenza is still a major potential threat of pandemics, and this work could indeed help the design of antiviral inhibitors targeting Flupol-PolII interfaces.

Major issues.

We thank the reviewer for taking the time to carefully evaluate our manuscript and to provide feedback.

1. As mentioned above, one main issue is the reliability/validity of the so called "post-cleavage" state given that it is derived from a very minor subpopulation and from only one fraction 14 in ED Fig. 5a.

We agree with the reviewer that only a small fraction of the initially picked particles were included in the final high-resolution reconstruction. We think there are two reasons for this:

1. This small fraction of particles ending up in the final reconstruction is a general problem for cryo-EM samples containing multiple components. This can be explained by compositional and conformational heterogeneity, as not all Pol II molecules would have FluPol bound. Also, FluPol may be bound in other conformations, which may be lost during particle sorting and averaging. Despite the above limitations,

the final reconstruction represents a stable intermediate state that is distinct from the pre-cleavage structure.

2. Here, we have deliberately used conditions where FluPol will be active and which results in RNA cleavage. RNA cleavage may lead to additional conformation changes and even FluPol dissociation. However, we resolved an intermediate complex, where FluPol has already cleaved the RNA but has not dissociated from Pol II, leading to a comparatively low abundance of FluPol on Pol II.

We chose the peak fraction to proceed further because it had the strongest overlap of fluorescent signals from Pol II (Cy5 label on template DNA) and FluPol (Alexa532 on 3'-vRNA).

2. Likewise, ED Fig. 1d shows that with 3.0 mM Mg, the major band remains the 35 nt capped RNA (as in the 0.1 mM fraction), and only a very minor amount of RNA is cleaved into two fragments (left lane).

The gel shown in the original manuscript originates from an experiment using the same concentrations of RNA and protein as in the analytical endonuclease assays. We have now redone this assay with the protein concentration used in complex assembly for cryo-EM on a small scale. We followed the protocol for the cryo-EM assembly until the point where we would have loaded the sample on the gradient and analyzed it using denaturing urea-PAGE instead. We replaced the previous gel with a gel from this experiment as it represents the biochemical state of the cryo-EM sample much better (**ED Figure 1e**). Here we observe that the majority of RNA had been cleaved in this sample. The additional time needed for gradient purification, dialysis and grid preparation may increase the amount of cleaved RNA in the EM sample even further. Therefore, we conclude that it is valid to assume that the majority of RNA is cleaved in the EM sample. Furthermore, we observe that the majority of the Pol II bound RNA after cleavage by FluPol has a length of ~24nt, with only a small proportion belonging to the longer ~31 nt RNA previously observed.

3. It would have been useful to try to analyze the identity of these fragments and correlate with what is observed in the postcleavage structure. Hence there is the possibility that what is called here the postcleavage state is only some intermediate that follows immediately the precleavage state. These should be at least discussed as possible limitations of the study.

We thank the reviewer for spotting the secondary cleavage product. As mentioned under 2., under the sample preparation conditions used for cryo-EM, the additional ~31 nt cleavage product only appears in a minor population and will likely be averaged out during data processing. We can only speculate why H7N9 FluPol cleaves at this position. This cleavage product is more frequent at lower concentrations of FluPol and RNA, and it seems specific to H7N9, as Influenza B/Memphis FluPol (Reviewer Figure 1) does not show this secondary product. Lengthwise, it would correspond to a second potential AG cleavage site, which might be recognized by the endonuclease without cap-binding by PB2.

We appreciate the reviewer's feedback on our assignment of the structure obtained relative to the FluPol transcription. First, we observe a majority of RNA at an RNA length of ~24 nt, in line with our post-cleavage structure. Second, if we had had another intermediate (either the minor ~31nt long cleavage product or something between pre- and post-cleavage), we would have cryo-EM density 5' of the cleavage site (**ED Figure 7f**). As an approximation, we have also extracted the post-cleavage cryo-EM density around the RNA path of the pre-cleavage complex (Reviewer Figure 3). We do not observe cryo-EM density at a reasonable threshold. Consequently, we do not think that what we observe is an intermediate between pre- and post-cleavage. However, we cannot completely rule out that a minor population of particles are already undergoing initiation. There is a little remaining density at the 3'-end of the capped RNA fragment, which is too poor to be modelled but is most likely the disordered RNA nucleotides of the capped RNA.

Reviewer Figure 3: **Post-cleavage cryo-EM density around the pre-cleavage RNA path.** Cryo-EM density for the RNA within FluPol between the PB2 cap-binding and PA endonuclease domains. The density is colored by thresholds. Red corresponds 0.6 and purple to 0.3 (same thresholds as in ED Figure 8b). The post-cleavage density was restricted to areas close to the pre-cleavage RNA in ChimeraX. The RNA is colored by heteroatom.

- The quality of the Figures should be markedly improved (eg Fig 2 and 4). Magnified views of capped RNA should be given and in connection with the possible criticism raised in 1-, the orientation of capped RNA in both the precleavage and even more in the postcleavage should be documented and supported by clear electron density maps documenting the proper assignment of bases.

We thank the reviewer for pointing out that the figures showing the Coulomb potential density map for the RNA can be improved. We included now panels showing only the RNA and the density, color-coded by the contour level used and, thereby, also showing the rigidity at the different positions in the RNA. We have done this for both pre- and post-cleavage. In the pre-cleavage state, we included the RNA residues in the active site and the corresponding density for comparison (ED Figure 3g,7e).

- In this respect having a 1D sequence of capped RNA clearly highlighting (with color code) what is built, what is disordered and the interactions with the proteins would help. This could be a modification of ED Fig 3.

We have included the requested schematics of the RNA and the interacting protein residues; RNA residues are color-coded in the same colors as used for contouring the density (ED Figure 3i,7h)

- 3-ED Fig. 3. Poor figure quality. Color choice? Comparison between the present structure of flupol-polIII and prior CTD-polIII complex (ref 2, 42) needs to be shown better. Give r.m.s deviations.

We agree with the reviewer that the CTD was difficult to see. We have removed the surface of FluPol for more clarity. We have calculated the requested RMSDs and included them in ED Figure 3.

- 4-In addition, the sentence line 246 “density discontinues abruptly” should be accompanied by electron density to document whether this break matches with the endonuclease active site and in addition temperature factors of Mg ions should be given (they should be lower for the postcleavage compared to precleavage?).

We have included the requested comparison of the densities in pre- and post-cleavage in ED Figure 7f. A comparison of B-factors of the Mg ion is, however, technically not very informative as pre- and post-cleavage maps differ in resolution, map sharpening procedure, particle numbers and normalizations (Relion vs. cryoSPARC).

- 5- Some of the writing could be improved/be more specific eg line 179, “phosphorylation of serine residues” which one?

In our assays we use CDK7, which is part of the CAK. This kinase has been reported to phosphorylate mainly serine 5, but also serine 7 *in vitro* (Glover Cutter *et al.*, Linhartova *et al.*). Since we have not biochemically determined the exact phosphorylation state of the CTD in our assays, we suggest to retain in the biochemistry section in this more general description. For our modelling, we were guided by the existing literature about the binding preferences of FluPol to serine 5 phosphorylated CTD. We do observe the phosphorylation of serine 5

and no additional cryo-EM density for phosphorylation on serine 2 or serine 7. Therefore, we changed the phrasing in the description of the EM structure to be more specific about the nature of CTD phosphorylation.

9. Line 176: “an interaction as proposed before” not very clear what was proposed.

We have altered the text to improve readability.

10. 6-For the mutation experiments in Fig. 3, given the uncertainty, can the authors envision validating their hypothesis (lines 207,208 key salt bridge between E141 and K627 from DSIF) which has a 8-10 fold impact on FluPol activity by swapping the charges to restore activity eg by a E141K, K627E double mutant and check on the minigenome assay?

We have further investigated the biochemical nature of this interaction using charge reversal mutants in the mini-genome context (Figure 3d). Furthermore, we have mutated the corresponding lysine in DSIF to alanine and have performed an endonuclease assay *in vitro* (ED Figure 3j,l). It seems that the nature of this interaction is more complicated than expected. We have adjusted our description of the interaction accordingly.

Other issues:

11. 1-Flupol mutant E119D is used throughout but its precise location is never shown in the structures including the Endonuclease domain... (unless I missed that).

We have not shown the precise location of the PA E119D mutation. It is a Mg chelating mutation, and we constantly show the Mg ion in our figure to orient for the endonuclease active site. This mutation has been used previously to study the endonuclease domain (see Kumar *et al.* 2021). The endonuclease domain architecture has been studied extensively. We included a figure panel in ED Fig 3i showing the endonuclease active site to illustrate the location of the mutation.

12. 2-Line 97: 96% identity: Vague formulation! Presenting 96% sequence identity? Pol II is a protein complex...

We specified this statement.

13. 3-Line 115: nearly no effect

We changed the wording for enhanced clarity.

14. 4-Line 129-130: The authors should consider to include a negative control (e.g. RNA + Pol II + CAK + DSIF) to confirm that the increased activity is dependent on FluPol and exclude the possibility that the 2-fold increase of RNA cleavage observed in the presence of the elongation factor DSIF is not caused by any carry-over of RNase contamination from the DSIF sample itself.

We agree with the reviewer that this important control strengthens the quality of the endonuclease assay. We repeated the endonuclease assay, including this control and have updated **Figure 1d** and **ED Figure 1b** accordingly.

15. 5-Line 132-133: The authors should provide a quantitative analysis of the signal intensity from ED Figure 1c for a better interpretation of the data.

Please see response to reviewer #1, point 8 and Reviewer Figure 2.

16. Actually from the data in ED Figure 1b and 1c, it is not clear whether the RNA extension activity can be correlated with the activity of the endonuclease activity.

Please see reviewer #1, point 8 for a detailed explanation and quantification.

17. 6-Fig 1: Swap data in a. and b. panels. A possibility is to present first in a. the SEC data and secondly confirm the presence of the protein subunits by Western-blot in b.

We have swapped Figure panels 1a and b as requested.

18. 7- Line 364-365: A paired parametric t-test is incorrect. The authors should perform an UNPAIRED parametric t-test when they want to compare the statistical significance between two conditions (unpaired conditions).

A paired parametric t-test is used when two measurements are conducted onto the same object before and after an experimental treatment. E.g. measure the body temperature before and after paracetamol intake.

However, in the present case the authors apply two different treatments (complex 3 vs complex 4) on their labelled RNA. So there is no connections, the t-test should be unpaired.

Reviewer Figure 1: Endonuclease cleavage rates in dependence of the substrate, lines connect data points from one replicate.

We agree with the reviewer that this experimental setup is not suitable for a paired t-test and we reassessed the statistical methods for analyzing the data. In addition to a consistent trend across the different substrates, we also observe variation in the absolute value between replicates (Reviewer Figure 3). Unfortunately, we cannot explain why the different replicates vary, but for sure, this is a source of systematic variation that is independent of the substrate used. To account for this, we changed our statistical analysis and now use a linear mixed-effects model, with condition/substrate as fixed effect and day/experiment as random effect (see Methods for details). Overall, the results agree well with those obtained using the paired t-test before.

19. 8-ED Table 1: precleavage B factor nucleotides 1117.71 A2, I hope it is a typo...

We thank the reviewer for spotting this typo; we corrected it.